# 3D printing with a 3D printed digital material filament for programming functional gradients

Sang-Joon Ahn[1,2], Howon Lee [2] ✉ & Kyu-Jin Cho [1,2] ✉

Additive manufacturing, or 3D printing attracts growing attention as a promising method for creating functionally graded materials. Fused deposition modeling (FDM) is widely available, but due to its simple process, creating spatial gradation of diverse properties using FDM is challenging. Here, we present a 3D printed digital material filament that is structured towards 3D printing of functional gradients, utilizing only a readily available FDM printer and filaments. The DM filament consists of multiple base materials combined with specific concentrations and distributions, which are FDM printed. When the DM filament is supplied to the same printer, its constituent materials are homogeneously blended during extrusion, resulting in the desired properties in the final structure. This enables spatial programming of material properties in extreme variations, including mechanical strength, electrical conductivity, and color, which are otherwise impossible to achieve with traditional FDMs. Our approach can be readily adopted to any standard FDM printer, enabling low-cost production of functional gradients.

Gradient in the local material composition, structure, and function is ubiquitous in nature[1–6]. Many biological material systems, such as fish scales[2,3], byssus thread[4], and bone[5,6], have been found to take advantage of this non-uniform characteristic to achieve remarkable properties[1,3]. Inspired by this, functionally graded materials (FGMs) have been proposed for various engineering applications in bioengineering[7,8], optoelectronics[9], and soft robotics[10–12]. Since FGMs are designed to have continuous spatial variation of properties, manufacturing has been a major challenge for the broader implementation of FGMs in engineering systems[3,13]. Additive manufacturing, or three-dimensional (3D) printing provides new opportunities for processing a wide range of materials from polymers to metal alloys to create FGMs within complex 3D geometries that would otherwise be inaccessible using traditional manufacturing[8,14,15].

Spatial control of material composition and distribution is essential in 3D printing of FGMs to address a broad range of property domains with a limited number of feedstocks. A widely known approach is a "digital material" printed with material jetting (MJ) 3D printing technique[16]. MJ process utilizes multiple inkjet printheads to simultaneously deposits numerous ink droplets. A wide range of mechanical properties and its spatial gradient can be easily produced by printing a mixture of inks for rigid and soft materials in specific concentrations[10,16–18]. However, MJ has been limited in its ability to create functional gradients in other property domains due to rheological constraints of the printing inks. Direct ink writing (DIW) systems can address a diverse set of functional inks, but each nozzle in DIW can extrude only a couple of inks with pre-determined composition. Although innovative studies have demonstrated remarkably detailed multimaterial structures by utilizing switching nozzles[19–22] or multi-nozzle arrays[23–25], continuous material change in DIW is still challenging for designing functional gradients. Digital light processing (DLP) 3D printing method has an advantage to create a stiffness gradient in a printed part using a grayscale digital mask[26–28], but it is limited to production of a monolithic part. Multi-material DLP printing is possible by employing multiple resin vats[29] or dynamic fluidic control[30], however, frequent material changes to create a material gradient produce excessive material waste. Moreover, aforementioned multi-material 3D printing approaches lack scalability because they often

[1]Soft Robotics Research Center, Seoul National University, Seoul, Republic of Korea. [2]Department of Mechanical Engineering, Institute of Advanced Machines and Design, Seoul National University, Seoul, Republic of Korea. ✉e-mail: howon.lee@snu.ac.kr; kjcho@snu.ac.kr

require specialized printing equipment, complex process, or expensive materials.

Compared to other 3D printing methods, fused deposition modeling (FDM) has distinct advantages such as affordability and accessibility, as seen by the largest installed base of 3D printers worldwide (up to 60%)[31]. However, FDM printing of functional gradients remains challenging since typical FDMs lack the capability for precise spatial control of material composition. This limitation stems from the fact that the extrusion nozzle in FDM can process only a single material at a time in the form of a filament[32,33]. Some FDM printers have multiple nozzles[34,35], but it is difficult to achieve continuously varying properties by switching between nozzles. Although mixing nozzles[36–38] with multiple filament inlets have been proposed, the material property in the printed structure is inhomogeneous due to non-uniform mixing between highly viscous molten polymers. To overcome these challenges, researchers have demonstrated that multiple materials can be incorporated at the filament level. FDM filaments with heterogeneous material composition have been created by extrusion[39] and thermal drawing[40,41], but they still have a longitudinally constant property due to the nature of the production methods. Different filaments have been connected via welding[42], yet only with a relatively simple spatial control over material composition. Recent works reported the production of a multimaterial filament through 3D printing[43,44], but due to the lack of systematic material design in the filament, the printed materials remained as a composite with separate material phases. As a result, not only did they show marginal improvement in mechanical properties, but they are also not capable of fabricating continuous gradient of material properties for FGMs.

Here, we present a 3D printed digital material filament (DM filament) for 3D printing of material gradient designs with widely tunable properties, using only a basic desktop FDM 3D printer and readily available materials. The direct construction of the DM filament through FDM enables precise control of the material composition and distribution, which leads to the desired functional gradients within a final object. We first show that a DM filament can be 3D printed in the same form factor of its commercial counterparts. Taking this a step further, we also demonstrate that multiple base materials can be joined in a DM filament with prescribed compositions and spatial arrangements. When the DM filament is subsequently fed back to the same printer, interdigitated layers of different materials in the DM filament are homogeneously blended as they pass through the heated extrusion nozzle. As a result, the properties of the printout are determined by the specific concentration of the raw materials in the DM filament. Therefore, the material composition and distribution encoded in the DM filament are translated to the spatial variation of the materials properties in the final 3D object. We call this process blended FDM (b-FDM) 3D printing. With b-FDM, it is possible to create 3D objects with the desired functional gradients over a variety of property domains, by combining only a few feedstock filaments. We show that b-FDM can be used to create a large range of mechanical stretchability, strength, and electrical conductivity. A rich expression of the full color spectrum can also be produced by b-FDM using only four filaments in primary colors. Taking advantage of this printing capability, we present a multifunctional origami gripper featuring rigid facets, flexible hinges, and integrated electric circuits including bending and tactile sensors, all of which were printed using a basic FDM printer with a single DM filament. The strength of this innovative approach lies in the fact that it can be readily applied to any standard FDM printers that are already in use across the globe. The b-FDM with a 3D printed DM filament may provide an easy and cost-effective way to manufacture multifunctional systems with material gradients.

## Results

### 3D Printing of DM filaments

FDM process creates 3D objects by extruding thermoplastic filaments as a thin extrudate and stacking them in a layer-by-layer fashion. We exploited this to directly print a cylindrical shape of the FDM filament. A 3D printed filament, or DM filament, is designed to have a standard filament diameter of 1.75 mm, so that it can be readily fed into any FDM 3D printer for the next round of printing (Fig. 1a, Supplementary Movie 1). With a layer thickness of 125 μm, we stacked a total of 14 layers to print a filament (Fig. 1b-d). Each layer is made up of 2 ~ 4 printing lines, or extrudates, with a width of 440 μm each. To print the multimaterial DM filament, we employed slightly adjusted printing parameters such as nozzle temperature and printing speed to meet the required conditions of the materials used (Supplementary Table 1). Compared to commercial filaments, DM filaments have a slight geometric discontinuity such as the stair-case effect and internal voids between extrudates, which should be compensated for by adjusting the rate of material extrusion. To test the compatibility of the DM filament, tensile test specimens and a 3D complex geometry (i.e., a vase) were printed using the same printer but with different types of filaments: a commercial filament and the DM filament (Supplementary Fig. 1). With a slightly higher (~5%) filament feed rate, both the DM filament and the commercial filament produced consistent results in terms of mechanical properties and dimensions (see Supplementary Information for details).

Unconventional printing capability further emerges when multiple filaments are used to print a DM filament. By switching base filaments and controlling printing sequence, multiple constituent materials can be heterogeneously integrated into a single DM filament with a precisely designed spatial arrangement. When a DM filament is supplied to a standard FDM printer, different materials interlaced across layers in the DM filament are blended while passing through the extrusion nozzle. Through this b-FDM process, the desired material properties are achieved in the printed object. As a proof-of-concept, we used a DM filament fabricated with only two base materials (standard filaments with different colors, cyan and yellow) to print a 3D object with 13 levels of color gradient (Fig. 1a). Our approach begins with two orthogonal user-defined information, namely, 3D geometry and color gradient. Based on this information, we established a filament programming scheme to encode desired material property distribution in the target 3D geometry (Supplementary Fig. 2 and 3, see Supplementary Information for details). Firstly, the target 3D geometry is parsed into a single line representing a printing nozzle path, mapping each point in the 3D geometry (voxel) onto a specific location on the DM filament (Supplementary Fig. 2a). Based on the material property (color gradient in this case) to be achieved at each voxel, the mixing ratio of two base materials at the corresponding location in the DM filament is defined (Supplementary Fig. 2b and c). The printing schedule to build the DM filament is optimized to minimize the number of material switch between base materials, considering the fact that 3D printed object is built layer-wise (Supplementary Fig. 2d and Supplementary Table 2). Consequently, the DM filament is encoded with the gradually changing material composition, which is then translated into the color gradient (Supplementary Fig. 2e and f). The 3D printed DM filament was then used in a FDM printer to print the target object. As a result, using the DM filament composed of only two base materials, 13 levels of color gradient were precisely distributed throughout a complex 3D geometry. The result showcases the potential of b-FDM to directly fabricate FGMs.

### b-FDM printing of digital materials with tailored properties

To achieve a target property through b-FDM, mixing of different polymers during extrusion must be considered in the DM filament design (Fig. 2). When a DM filament is extruded through a confined channel in the heating nozzle, a laminar flow develops due to the high

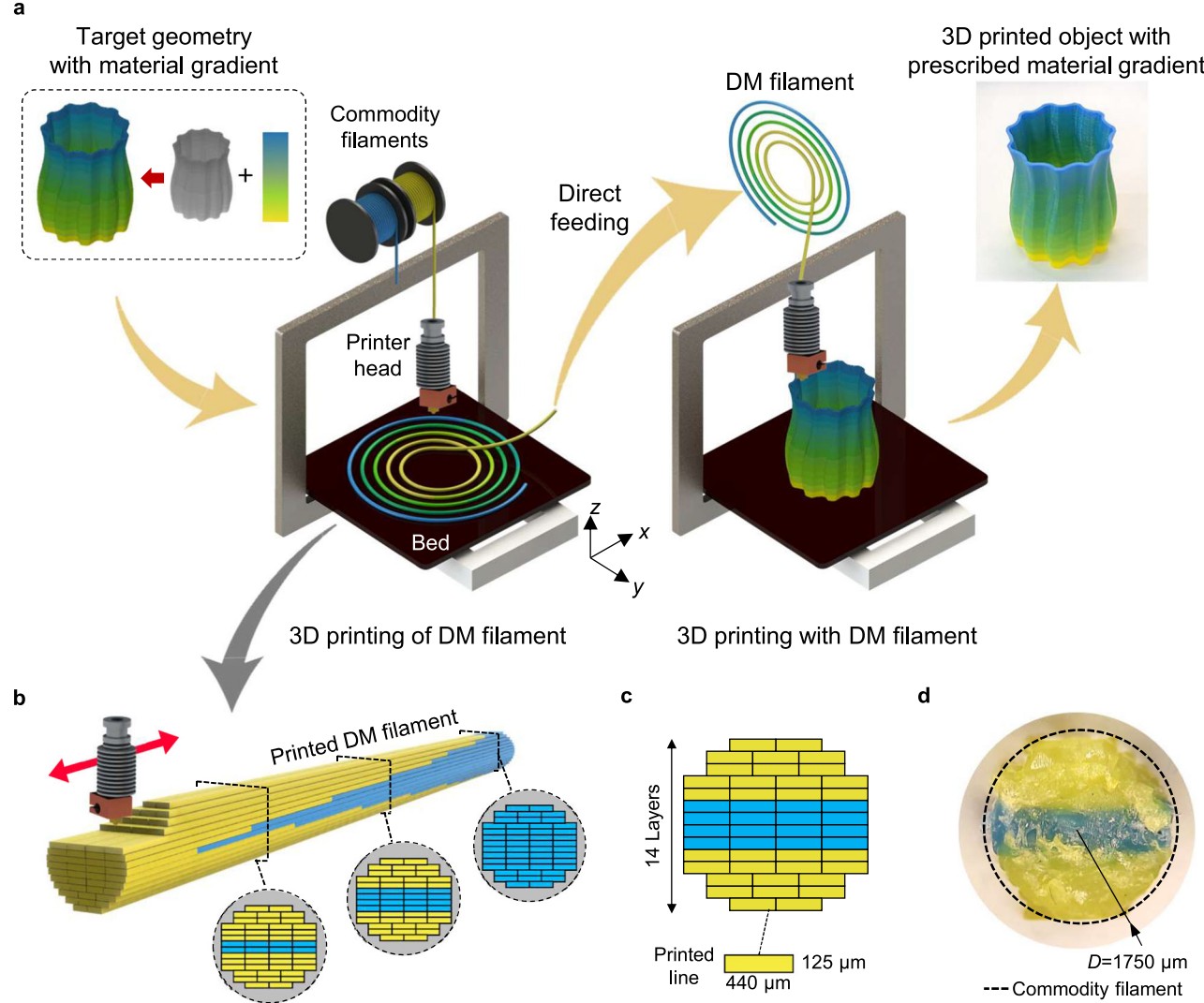

**Fig. 1 | b-FDM 3D printing with a DM filament. a** Schematic illustration of b-FDM printing process. A DM filament with a prescribed material composition is first 3D printed according to the desired material property distribution in the target 3D geometry (programming, encoding). The desired material gradient appears in the 3D object printed with a standard FDM printer. (decoding). **b** Constituent materials (two filaments with different colors, cyan and yellow) can be heterogeneously printed into a single DM filament with a precisely designed spatial arrangement. The red arrow indicates the nozzle movement direction during DM filament printing. **c-d** A DM filament is designed to have same form factor of commodity filaments with diameter of 1.75 mm, as shown in schematic and optical image of a cross-section.

viscosity of the molten polymer. Therefore, convective mixing between polymers hardly takes place, leaving polymer interdiffusion as a primary mechanism for mixing[38,45,46] (Fig. 2a). However, the mixed zone in the extrudate is expected to be extremely small, resulting in locally aggregated materials in the final object (see Methods for calculations). This issue can be addressed through the design of the DM filament in which different materials are deposited in an alternating fashion. Increased interface between different materials facilitates homogeneous mixing during extrusion[47,48]. For a given interdiffusion depth, the mixed zone in the extrudate increases with the total interfacial width in the cross-section of a DM filament (Fig. 2b). To study the effect of the internal design of a DM filament on the mixing of constituent polymers, we introduce a dimensionless homogeneity parameter $\eta$, defined as

$$\eta = \sum W_i / d_f \qquad (1)$$

where $W_i$ and $d_f$ denote the interfacial width between layers of different materials and the diameter of the DM filament, respectively.

We first postulated that $\eta = 0$ for a single material where no material interface exists (Fig. 2b, i). For a DM filament with two base materials in a 1:1 ratio, $\eta$ increases with the number of interdigitated layers (Fig. 2b, ii-iii). Ideally, a DM filament with infinite number of interdigitated material layers ($\eta = \infty$) would result in complete mixing (Fig. 2b, iv). With two poly-lactic acid (PLA) filaments in different colors (cyan and magenta), we prepared a set of DM filaments with varying $\eta$ values ranging from 1 to 11, while maintaining a 1:1 ratio of the two materials (Fig. 2c). The smallest homogeneity parameter $\eta = 1$ is for the case where two materials are simply stacked without interdigitation. On the other hand, $\eta$ is maximum ($\eta = 11$) when the materials change layer by layer (13 times) in the DM filament. Different DM filament designs with $\eta = 3.5, 6.0,$ and $8.0$ were also printed, interdigitated layer designs of which necessitate 4, 7, and 9 times of manual material switching during the filament fabrication process, respectively. As clearly seen from the cross-sections of the extrudates (Fig. 2c, bottom), polymer mixing is significantly improved as $\eta$ increases. The boundaries between two colors are clearly visible until $\eta = 6.0$, but they become

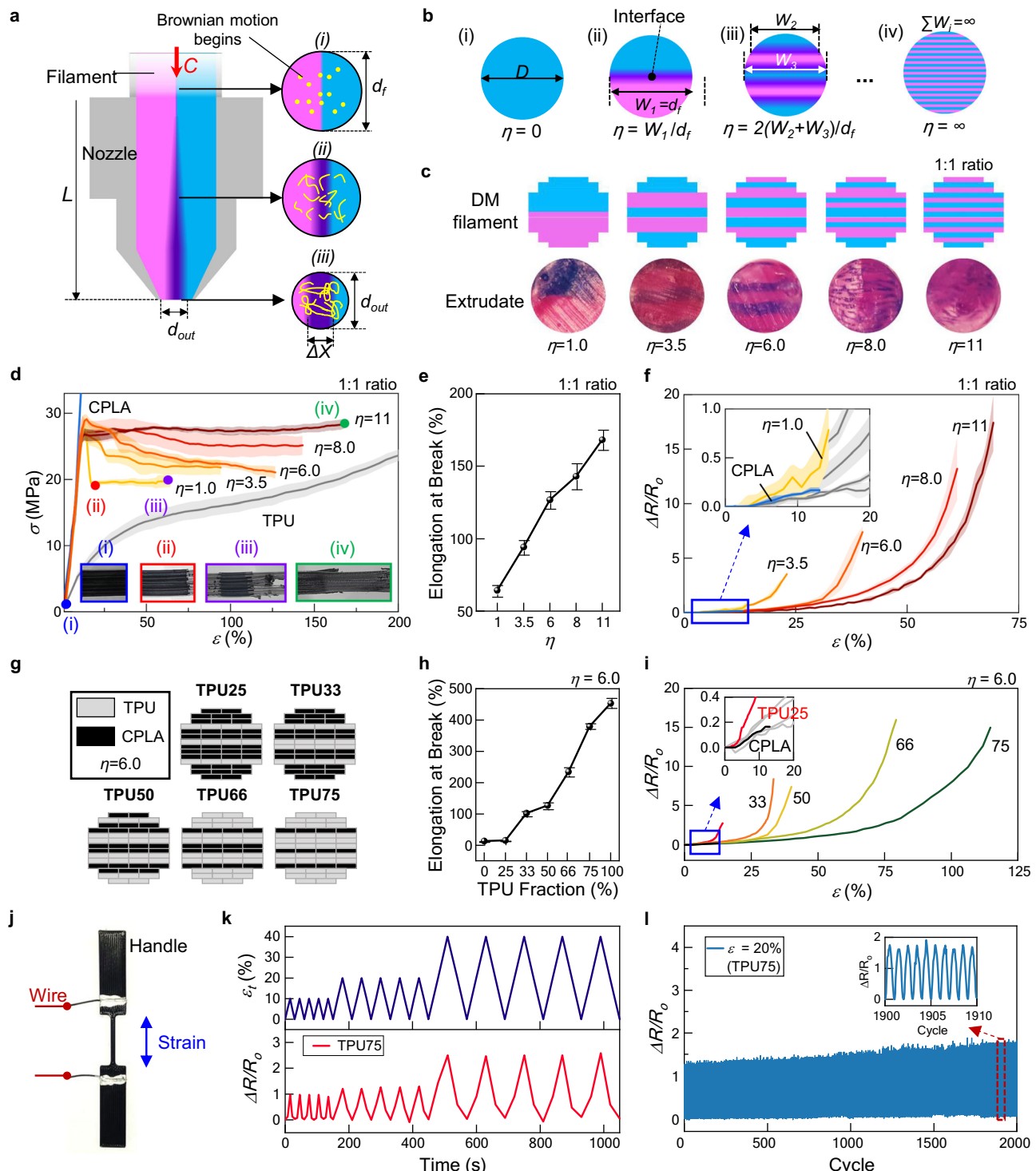

**Fig. 2 | b-FDM printed digital materials with tunable and multi-functional properties. a** Schematic of the extrusion of a DM filament with two materials (illustrated in cyan and magenta). While a DM filament (diameter $d_f$) is supplied at a feed rate of $C$ and passes through a heated region (length of $L$) in the nozzle (outlet diameter $d_{out}$), polymer interdiffusion (illustrated in purple) takes place across the material interface with a depth of $\Delta x$. **b** DM filament design with interdigitated multi-layer arrangements and corresponding homogeneity parameter $\eta$. **c** Schematic of different DM filament designs with varying $\eta$ (top). Optical images of extrudates (bottom) were obtained by extruding DM filaments through a heated nozzle without deposition on the bed. **d** Mechanical responses of CPLA-TPU blends printed using DM filaments with varying $\eta$. **e** Elongation at break as a function of $\eta$. All error bars represent the standard deviation ($n = 5$). **f** Electrical responses during

stretching of the specimens produced with DM filaments with varying $\eta$. **g** Designs of CPLA-TPU DM filaments with varying mixing ratios, while $\eta$ was kept constant at 6. Numbers indicate the volume fraction of TPU in each design. **h** Elongation at break as a function of TPU fraction. All error bars represent the standard deviation ($n = 5$). **i** Representative electrical resistance change with respect to the base line resistance, $\Delta R/R_0$, during stretching. **j** The strain sensor that is b-FDM printed using a TPU75 DM filament. The sensible gauge had overall dimensions of 20 mm × 1.6 mm × 1.2 mm, while handles attached to the grips were printed on both sides. **k** Electrical resistance variation during triangular strain cycles with the peak strains of 10%, 20% and 40%. **l** Electrical resistance variation during cyclic test with the peak strain of 20%, demonstrating a robust performance of the strain sensor.

less apparent when $\eta$ is greater than 8.0, indicating more homogeneous mixing of the two materials.

b-FDM creates the digital materials with hybrid characteristics by homogeneously blending multiple base materials combined in the DM filaments. For example, conductive PLA (CPLA) is stiff, brittle, and electrically conductive due to carbon black (CB) nanoparticles in it, whereas thermoplastic polyurethane (TPU) is soft, flexible, and non-conductive. By adding TPU to CPLA through b-FDM, electrically conductive and flexible materials can be created. To test this capability, we used a 1:1 ratio of CPLA and TPU to print an array of DM filaments with varying $\eta$. Mechanical properties of the specimens printed with CPLA-TPU DM filaments were examined by tensile testing (Fig. 2d, see Methods). The specimens with $\eta = 1$ showed delamination of CPLA from TPU matrix, followed by brittle fracture when strain reached 17% (Fig. 2d, ii and iii), indicative of incomplete mixing of CPLA and TPU. On the contrary, remarkable improvement in ductility was observed for $\eta = 11$ (Fig. 2d, iv), with a substantial increase in fracture strain up to 168% (Fig. 2e). With the same material composition, different levels of homogeneity resulted in dramatic differences in the mechanical behavior. Moreover, improved mixing of CPLA and TPU also helps to preserve electrical conductivity of the printed polymer during deformation. When stretched, the resistance of CPLA-TPU blends increases rapidly until the conductive CB networks within the polymer matrix break. Figure 2f shows the normalized resistance change $\Delta R/R_O$ (where $\Delta R = R_1 - R_O$, with $R_O$ and $R_1$ being the resistances in initial and deformed states, respectively) of the CPLA-TPU mixtures with different $\eta$. For $\eta = 1$, electrical conductivity was lost at a strain of 14%, showing marginal improvement compared to the pristine CPLA (Fig. 2f, inset). As $\eta$ gradually increased, electrical conductivity was maintained at higher strains up to 69%. Therefore, CPLA-TPU blends can be used for stretchable sensing applications. Similar results were also obtained for different CPLA-TPU mixing ratios (Supplementary Fig. 4). The results demonstrate that increasing $\eta$ facilitates effective blending of basic constituents, resulting in an enhanced performance of b-FDM printed digital materials. However, this increase in $\eta$ also requires more frequent switch of the feedstock throughout the fabrication of the DM filament, resulting in a time-consuming process (approximately 3 min for each change). Through our investigation, we have identified a suitable compromise in this trade-off at values of $\eta$ ranging from 6 to 8.

Furthermore, varying the concentration of the base materials in the DM filaments can significantly expand the range of achievable properties in the printed structure. We designed and printed a series of CPLA-TPU DM filaments while varying the volume fraction of TPU from 25% to 75%, denoted as TPU25, TPU33, TPU50, TPU66, and TPU75 (Fig. 2g). $\eta$ was kept constant at 6 for all DM filaments to ensure homogeneous mixing of CPLA and TPU. Uniaxial tensile tests were performed on b-FDM printed specimens, while mechanical and electrical responses were measured (Supplementary Fig. 5, see Methods). Results indicate that stretchability of printed structures strongly depends on material composition in the DM filament, varying over 30-fold from 13.9% (CPLA only) to 453.6% (TPU only) (Fig. 2h and Supplementary Fig. 5a). Electrical property of the printed digital materials also depends on the material composition of the DM filament. As the volume fraction of TPU increases, there is a corresponding decrease in the concentration of CB nanoparticles originating from CPLA within the DM filament. This reduction leads to a substantial rise in the baseline resistance, $R_O$, by two orders of magnitude from 135 to 19,314 Ohm (Supplementary Fig. 5b). In the meantime, the variation range of $\Delta R/R_O$ increased significantly due to improved stretchability (Fig. 2i). Similar results were also obtained under 3-point bending tests, showing a wide range of tunable properties of CPLA-TPU blends (Supplementary Fig. 6, see Methods). The results suggest that spatial design of material composition within a DM filament can be effectively utilized to tailor the material properties of 3D printed parts.

To further investigate multifunctional properties of b-FDM printed digital materials, we printed a strain sensor using a TPU75 DM filament, as shown in Fig. 2j (see Materials and Methods for detail). When triangular strain cycles with the peak strains of 10, 20, and 40% were applied, repeatable output signals were generated across the entire range of strain (Fig. 2k). Moreover, this stable response was maintained over 2000 cycles of 20% peak strain, demonstrating a robust performance of the strain sensor (Fig. 2l). A marginal increase in peak resistance was observed, potentially attributed to viscoelasticity-induced residual strain in the polymer matrix.

## Functionally graded materials programmed by DM filaments

The b-FDM printing with DM filaments offers unprecedented capability to precisely control functional gradients in multiple property domains (Fig. 3). For example, weak interfacial bonding in multi-material 3D printing is a critical challenge as it affects the integrity of printed objects. Although creating a mechanical gradient at the material interface has been reported to improve interfacial bonding[1–8,21], conventional FDM is limited in this capability due to its simple material processing procedure. Here, we demonstrate enhanced interfacial bonding by printing a material gradient using only one DM filament. We used PLA and polyethylene terephthalate-G (PETG) as base materials for the DM filament because they have similar mechanical properties, but do not adhere each other. In the tensile test specimens, we positioned the two materials in a symmetric arrangement, with PLA (illustrated in red) at the center and PETG (illustrated in gray) at both ends, as shown in Fig. 3a and b (See Methods for details). $x$ represents the relative distance from the center, ranging from 0 to 1. All specimens were printed with the nozzle path in the transverse direction (or perpendicular to the specimen's longitudinal direction, i.e., $x$-axis), since FDM printed objects are most susceptible to forces applied in this direction (Fig. 3b). To investigate the effect of material gradient on the interfacial bonding, we prepared three PLA-PETG DM filaments to create different profiles of material gradient. The volume fraction of PETG was changed with stepwise variations of 50%, 16%, and 8%, resulting in 1-, 5-, and 11-step gradient profiles within the test specimens, respectively (Fig. 3c, bottom). The interdigitated layer arrangements in the DM filaments were designed to ensure uniform mixing of PLA and PETG during b-FDM process (Supplementary Fig. 7). In addition, to serve as a control experiment, a specimen with a sharp material change (nongraded) was also printed (see Methods). Fracture stresses and corresponding fracture locations obtained from tensile tests are shown in Fig. 3c. As the number of gradient steps increased, there was a three-fold enhancement in the average fracture stress ($\sigma_{\text{fracture, avg}}$) from 9.92 MPa for nongraded to 30.16 MPa for 11-step. In addition, increasingly widespread distribution of fraction locations across the specimen was observed as compositions changed more gradually It is noteworthy that there is a trade-off between the steepness of the material gradient and the band size of the material interface. A more abrupt change in the material composition (i.e., 1-step or 5-step profile) would result in a more distinctive, narrower interface; however, it may also lead to a loss of the characteristics of FGM, resulting in a limited enhancement of bonding strength. This result suggests that b-FDM-enabled material gradient programming can facilitate seamless multimaterial 3D printing and promote robust bonding between different materials with mechanically invisible material interfaces.

b-FDM printing also enables the creation of diverse color gradients within 3D geometries. Achieving full color 3D printing has been limited to specialized 3D printers with costly materials. Gradient programming with a DM filament provides a cost-effective solution for generating a rich color spectrum using only a few standard filaments. We demonstrate the successful printing of 36 colors within a 3D object, achieved by employing a single DM filament composed of only four base materials (Fig. 3d–f and Supplementary Fig. 8a). In the upper

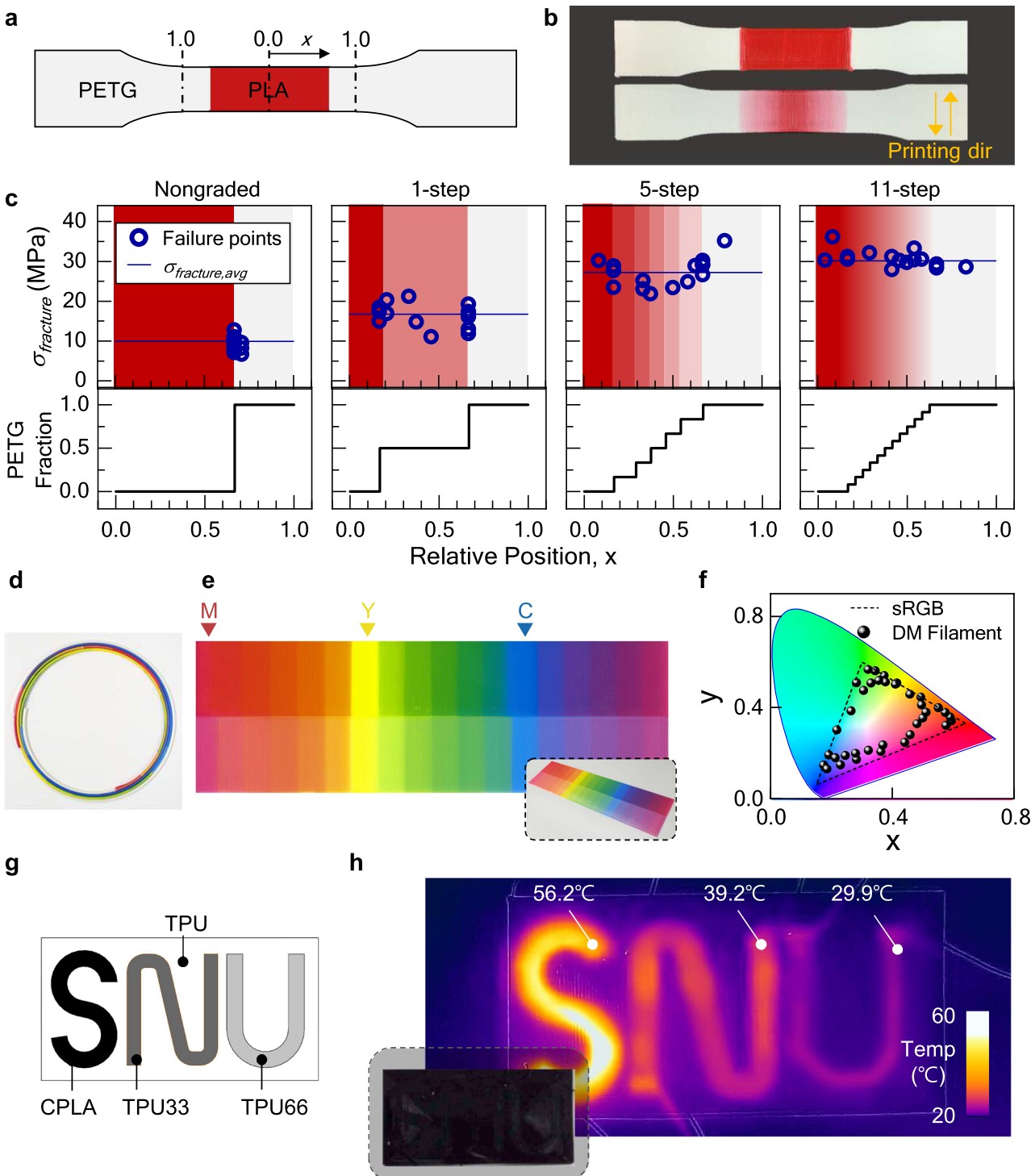

**Fig. 3 | b-FDM 3D printed functional gradients. a** Design of a tensile test specimen with PLA (illustrated in red) and PETG (illustrated in light gray). The relative distance from the center is represented as x, ranging from 0 to 1. **b** Printed tensile test specimens with different material gradients at the interface. Printing direction was perpendicular to the longitudinal axis. **c** Fracture stresses and the corresponding locations of fracture from the specimens with different material gradient designs across the interface. The blue lines indicate the mean value of the fracture stresses. **d** Printed DM filament for full color 3D printing. **e** b-FDM printed planar object (inset) with 36 colors. The upper half of the object shows graded colors between any two of the primary colors (cyan, magenta, and yellow). The lower half shows the printed colors with increased brightness. **f** Resulting color spectrum mapped on CIE diagram. The black dotted triangle indicates the standard RGB (sRGB) gamut. **g** Schematic of a multimaterial device programmed with different levels of electrical resistance. **h** b-FDM printed result (inset) and an infrared photograph which reveals each letter with thermal gradient.

half, we generated 5 graded colors between each pair of primary colors (cyan, magenta, and yellow) by incrementally changing their concentrations by 16%. Furthermore, in the lower half, we demonstrated increased brightness for each color by adding 50% of a white filament. We mapped the generated color spectrum on the Commission internationale de l'éclairage (CIE) map, as shown in Fig. 3f (see Methods for details). The result shows a broad range of colors achievable through b-FDM, which is comparable to standard RGB (sRGB) gamut (illustrated in a dotted triangle) observed in digital devices. The significance of the result lies in the fact that it was achieved by using only a basic FDM printer and low-cost materials.

The sample principle of designing gradients can be further extended to the programming of electrical properties (Fig. 3g, h). In the previous section, we showed the modulation of the electrical resistance of a printed part by controlling the material compositions within CPLA-TPU DM filaments. Based on this, we designed a DM filament with distinct segments having different electrical resistance, 3.26, 7.81, and 25.3 kΩ, respectively (Supplementary Fig. 8b). Moreover, we ensured that the longitudinal position of each segment in the DM filament was synchronized with the printing sequence, so that each segment was printed in the correct order and accurately mapped onto the specific printing pattern (Supplementary Fig. 8c). Specifically, the letters 'S', 'N', and 'U' were printed with CPLA, TPU33, and TPU66, respectively, where the number denotes the concentration of TPU (Fig. 3g). The substrate was printed with pristine TPU for electrical insulation. Due to the presence of embedded CB nanoparticles, the printed letters are visually indistinguishable because CPLA, TPU33, and TPU66 all appear black in color (Fig. 3h, inset). However, the application of a DC voltage of 85 V in parallel induces a different level of Joule heating power, $P$, of 2.21, 0.93, and 0.29 W, respectively (based on $P = V^2/R$, where $V$ and $R$ denote the input voltage and the resistance, respectively). As a result, three letters were heated above room temperature (25 °C) with the increase proportional to the supplied power (31.2, 14.2, and 4.9 °C, respectively), making the invisible letters appear in the infrared image (Fig. 3h, see Materials and Methods).

## All-in-one printing of a multifunctional origami gripper
To highlight the capability of the b-FDM method, we fabricated a multifunctional origami gripper incorporating rigid facets, soft hinges, and integrated electrical components including bending and tactile sensors (Fig. 4a). Notably, the entire device was printed in a single printing process using a single DM filament. Since the gripper was designed based on the Miura-Ori[35,49] folding pattern, it can be easily printed as a planar shape (Fig. 4b), but exhibits out-of-plane deformation when folded along the prescribed creases (Fig. 4c). To accomplish this large folding without mechanical failure, it is crucial to establish robust interfacial bonding between the rigid facets made of stiff PLA and the flexible hinges made of soft TPU. Furthermore, we also incorporated integrated electrical components in the gripper including folding and tactile sensors (Fig. 4a). To ensure accurate origami folding detection, the folding sensor should possess high sensitivity to bending, while the printed conductive path remain insensitive to such bending motion. Also, to detect the gripping of objects with arbitrary shapes, the tactile sensors located at both ends of the gripper should possess sufficient flexibility to easily conform to the contours of any held object. Such a wide range of functional requirements across various property domains is challenging to achieve with existing 3D printing methods, especially with conventional FDM.

Using our DM filament design strategy, we programed the required mechanical and electrical gradients within the gripper, enabling all-in-one printing of the origami gripper (Fig. 4d). The flexible hinges were printed using TPU with the lowest flexural modulus ($E_b$) of 73.3 MPa, while stiff PLA ($E_b$ of 1598.5 MPa) was used for the facets. To ensure strong bonding between these two mechanically distinct materials, we implemented a material gradient using the 5-step gradient profile, which exhibited a sufficient interfacial bonding as shown in Fig. 3c. As we gradually decreased the fraction of TPU, $E_b$ in the PLA-TPU blends increased gradually, effectively reducing the significant stiffness mismatch between the facets and the hinges (Fig. 4d). The electrical properties were also tailored by adjusting TPU concentration in the CPLA-TPU mixture, as we demonstrated in Fig. 2. To achieve high sensitivity in folding, we employed TPU25 (TPU fraction of 25%) exhibiting the highest gauge factor (GF = $(\Delta R/R_O)\varepsilon$, where $\varepsilon$ denotes applied strain) of 3.69 under flexural deformation (Supplementary Fig. 6). The conductive path, which should be insensitive to fold, was printed using TPU50 having the lowest GF of 1.09. Since conformability is important for the tactile sensor, we utilized TPU75 due to its lowest $E_b$ of 387.7 MPa among the conductive blends. The individual components of the gripper and their corresponding digital materials were encoded onto specific locations on the DM filament, considering the printing sequence of the whole device. Specifically, the filament was programmed to create the flexible hinges as a substrate (Fig. 4d, i), the electrical components, the mechanical gradient, and the rigid facets, in sequential order across the layers (Fig. 4d, ii-iv). Meanwhile, for comparison, we replicated the gripper by manually switching pristine TPU, CPLA, and PLA filaments to create the hinges, the electrical components, and the rigid facets, respectively.

As a result, the b-FDM printed origami gripper displays sharp folding along the creases and grasps object without any mechanical failure (Fig. 4e). In contrast, the conventional FDM-printed origami gripper without material gradient exhibited severe delamination as soon as it was folded (Fig. 4f, Supplementary Movie 2). Due to the integrated sensors, the gripper is also capable of detecting both folding and contact (Fig. 4g, h, Supplementary Movie 3). The folding sensor and tactile sensors were individually connected to the ohmmeter in order to characterize the responses with changes in resistance, $\Delta R/R_O$ (Supplementary Fig. 9)[38,50–53]. Throughout the folding deformation, the folding sensor exhibited a consistent response with a $\Delta R/R_O$ up to 10%, while the undesired signal from tactile sensors and the conductive path remained small, measuring no greater than 2% (Fig. 4h, i). Upon contact, the tactile sensors conformed to the object to provide a secure grip and produced a noticeable change in the signal (Fig. 4h, ii). This result highlights the remarkable capability of b-FDM printing to seamlessly incorporate a wide range of desired functionalities into engineering systems through the implementation of material gradient designs.

## Discussion
The manufacturing of FGMs with precise spatial variation of materials properties is still a formidable challenge. In this study, we present 3D printed multimaterial filaments for the direct 3D printing of objects with desired functional gradients across various property domains, such as mechanical strength, electrical conductivity, and color. Our study demonstrates successful encoding of desired properties and their 3D spatial distribution into a multimaterial DM filament. Decoding of this information through the b-FDM process enables physical realization of both the intended geometry and the desired property distribution in the final printout. To demonstrate the unique advantages of our method, we presented various applications, including strain sensor, multicolor printout, electrical resistance distributions, and functional origami gripper, all of which are beyond the capabilities of conventional FDMs. It is noteworthy that this approach enables the production of an extensive range of material properties using only a limited number of commercially available low-cost filaments. Hence, the b-FDM with 3D printed DM filaments offers a new pathway to unleash the full potential of FGMs for various engineering application. However, the challenges may arise when attempting to program various material properties within a narrow range, primarily due to the

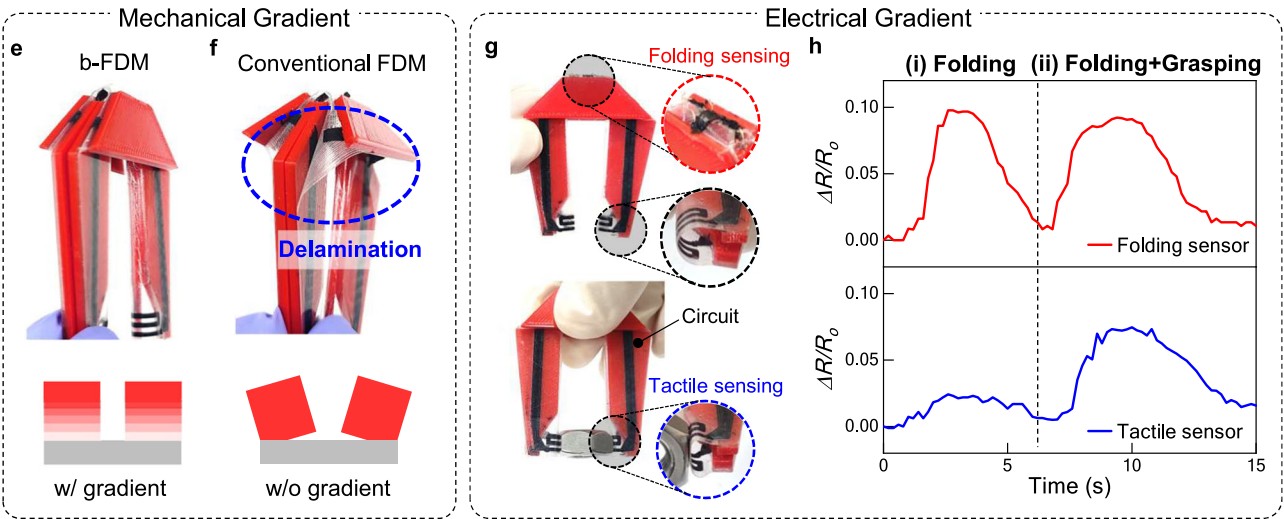

**Fig. 4 | b-FDM 3D printed multifunctional origami gripper. a** Schematic illustration of the multimaterial, multifunctional origami gripper. **b** As-printed origami gripper. **c** The folding sequence of the gripper. **d** DM filament design for b-FDM printing of the gripper with mechanical and electrical gradient. A specific material composition ($\phi$) and homogeneity parameter ($\eta$) to program desired material properties (illustrated with blue shade) such as flexural modulus ($E_b$) and gauge factor (GF) are provided in the table. **e**, **f** The effect of the mechanical gradient on the integration of heterogeneous properties. The folding behavior of **e** the b-FDM printed origami gripper is contrasted with **f** the conventional FDM printed gripper which exhibits clear delamination at the interface. **g**, **h** The effect of electrical gradient on the integrated circuit. **g** Folding and gripping detection mechanism based on electrical gradient design. **h** Resistance variation of the folding sensor and the tactile sensor during the sequential folding and contact.

limited printing resolution of the standard FDM[54]. Consequently, achieving functional gradients through b-FDM programming may require a wide material interface. In addition, there is also room for improvement in optimizing printing time for DM filaments and extending the length of a printable DM filament in a single printing. Nevertheless, it is worth highlighting that the presented approach holds universal applicability, as the DM filaments can be 3D printed and utilized with any FDM 3D printer (Supplementary Movie 4 and Supplementary Fig. 10). Thus, we envision that the presented method can readily serve as a powerful add-on to existing FDM printers that are already in use worldwide, pushing the boundaries of what is achievable in 3D printing.

## Methods

### Materials and apparatus setup

The DM filaments were 3D printed using an off-the-shelf FDM 3D printer (Original Prusa i3 MK3S, Prusa Research) with a 0.4 mm-diameter nozzle and 1.75 mm-diameter feedstock inlet. Zortrax M300 (Zortrax, Poland) and Cubicon single plus (Cubicon, South Korea) were used to show that our DM filaments can also be used in different FDM printers and yet produce the same result (Supplementary Fig. 10). Four types of commercially available base filaments were obtained: (i) PLA filaments were obtained from Prusa Polymers, Prague, Czech; (ii) PETG filaments were obtained from Prusa Polymers; (iii) CPLA filaments composed of embedded carbon black nanoparticles in PLA polymer matrix were obtained from Protoplant, Inc, Vancouver, Canada; and (iv) TPU filaments with maximum tensile strain of 400% were obtained from eSUN Industrial Co., Ltd., Shenzhen, China. The printing parameters including the temperature and the printing speed were adjusted for stable printing (Supplementary Tables 1 and 2). To ensure good adhesion, the print bed was heated to 70 °C, and the nozzle temperature was changed from 220 °C to 240 °C, following the manufacture's recommendation. When extruding DM filaments, the nozzle temperature was set to be the maximum allowable temperature to ensure sufficient material supply. For instance, we used the nozzle temperature of 220 and 240 °C accordingly to build the PLA-PETG DM filaments shown in Fig. 3, but the temperature was fixed to 240 °C while 3D printing the final specimen. Other parameters including the printing time of each DM filament and final object can be found in Supplementary Table 2.

### Estimation of the polymer interdiffusion depth

We estimated the interdiffusion depth across the interface of the molten polymers, $\Delta x$, based on the Fickian diffusion. Supposed that a DM filament with a diameter $d_f$ consists of two materials, cyan and magenta, as illustrated in Fig. 2a. It is supplied at a feed rate of $C$ and passes through a heated region with a length of $L$ in the nozzle, resulting in a thin extrudate at the nozzle outlet with a diameter $d_{out}$. Polymer interdiffusion occurs at the material interface as its temperature $T$ rises above the glass transition temperature, $T_g$ (Fig. 2a, ii). Based on the Fickian diffusion, the depth of diffusion across the interface can be expressed as[34,45,46]:

$$\Delta x = 2\sqrt{2Dt} \qquad (2)$$

where $\Delta x$, $D$, and $t$ denote the interdiffusion depth (illustrated in purple shade), the diffusion coefficient, and the extrusion time that is defined as $t = L/C$, respectively (Fig. 2a, iii).

To define the diffusion coefficient, we first adopted the formula[44] that relates interdiffusion coefficient ($D$) and the molecular weight ($M_w$) of the poly (methyl methacrylate) (PMMA), that is:

$$D = 38.5 * M_w^{-2.097} \qquad (3)$$

We chose this formula since molecular weight and viscosity of PMMAs are similar to those of the PLAs, and the thermal condition (190 °C) is also close to the 3D printing temperature (220–240 °C). We estimated $M_w$ of PLAs to be 1.474 g/mol E5 corresponding to PLA 4043D (NatureWorks LLC.), since PLA 4043D is widely used to fabricate FDM printable filaments[55]. Thus, the interdiffusion coefficient was calculated to be $D = 5.59 \times 10^{-10}$ cm²/s. Secondly, we calculated the extrusion time, $t$, from the printing condition of the DM filament. $L$ is fixed at 19.6 mm based on the hardware setup, while $C$ is set to be 0.1 mm/s to ensure sufficient extrusion time for the interdiffusion. By substituting given values in Eq. (2), the interdiffusion depth is estimated at 9.28 µm. We postulated that molecular behavior of other raw materials such as TPUs and CPLAs are similar to PLAs.

### Mechanical and electrical testing methods

Optical images of the cross-section of extruded materials were recorded using a microscope (Olympus SZ61). Tensile and flexural tests were conducted using a universal testing system (Instron) according to ASTM D638-14 and ASTM D790 standards, respectively. Specimens for the tensile test of the CPLA-TPU blends were designed according to the ASTM D638-Type 5. For the electrical measurements of the specimens, strain sensor, and origami gripper, we attached copper wires using silver paste ("Leitsilber" conductive silver content, Ted Pella Inc.) to connect specimens to a custom-made resistance measurement device based on Arduino. Infrared image was recorded and analyzed using an infrared camera (FLIR One Pro, Teledyne FLIR LLC).

### Experimental method for mechanically graded structures

Tensile tests were performed to characterize the functionally graded designs composed of PLAs and PETGs shown in Fig. 3. First, as a reference, uniform specimens consisting only of PLAs or PETGs were printed using commercially available filaments. The fracture stress of PLAs and PETGs were measured as $40 \pm 2.3$ MPa and $44 \pm 1.7$ MPa, respectively. As no standard exists for testing the multimaterial interface, the specimens were designed according to the ASTM D638-Type 1 with modifications as follows. The first materials, PLAs, were inserted in the gauge symmetrically with an overall length of 6 mm. The total gauge length was set to be 48 mm to contain the second materials, PETGs, with length of 8 mm, and material gradients at the interfaces with length of 10 mm placed at both sides, symmetrically. The gauge width was increased to 14.4 mm with the thickness of 3.2 mm. To create different gradient profiles, we first designed the DM filament with 11-step gradient profile (Supplementary Fig. 7). Different DM filament designs were selected to create other gradient profiles; specifically, 50/50 was used for 1-step profile, while 16/84, 35/65, 50/50, 65/35, 84/16 for 5-step profile.

### Experimental methods for full color printing

The color gradient specimen shown in Fig. 3 has dimensions of 180 mm (width) × 48 mm (length) × 0.4 mm (thickness). The specimen is divided into an array of 18 × 2 sections, with each section measuring 10 mm × 24 mm in size. 18 graded colors were arranged on the upper half, and their corresponding colors with increased brightness were placed on the bottom half. To characterize the result, the specimen was scanned and analyzed through a commercial scanner (HP OfficeJet Pro 9010) and ImageJ. The RGB values of each color section were calculated as the mean of the values measured across their respective sections. To plot the results on the CIE map, MATLAB function 'rgb2xyz' was used to convert RGB values into CIE 1931 xy color space. Also, 'makecform' and 'applycform' were used for RGB to CMYK transformation, where C, M, Y, and K denote cyan, magenta, yellow, and black, respectively (Supplementary Fig. 2f).

## Data availability

The authors declare that all relevant data supporting the findings of this study are available within the article and its Supplementary Information files. Source data are provided in this paper.

## Code availability

The source code used in this work is available at https://doi.org/10.5281/zenodo.10899879.

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

## Acknowledgements

We acknowledge the financial support from the National Research Foundation of Korea (NRF) Grant funded by the Korean Government (MSIT) (RS-2023-00208052, 2022R1A2C2003566, and RS-2023-00218543) and the Korea Medical Device Development Fund grant funded by the Korea government (the Ministry of Science and ICT, the Ministry of Trade, Industry and Energy, the Ministry of Health & Welfare, the Ministry of Food and Drug Safety) (Project Number: RS-2020-KD000175). H.L. also acknowledges the financial support from the SNU Creative-Pioneering Researchers Program.

## Author contributions

S.-J.A., H.L. and K.-J.C. conceived and designed the work. S.-J.A. and H.L. designed the experiments and conducted related data analysis and interpretation. S.-J.A. conducted the experiments and designed the b-FDM 3D printed structures. S.-J.A. and H.L. wrote the manuscript. H.L. and K.-J.C. edited the manuscript. All authors discussed the results.

## Competing interests

The authors declare no conflict of interest.
