## [Peer Review File · Nature Communications]

3D printing with a 3D printed digital material filament for programming functional gradientsREVIEWER COMMENTS

Reviewer #1 (Remarks to the Author):

This work is well written, the topic is interesting and worthy of publication. The authors show how to print gradient materials with an off the shelf generic FDM printer. This could have a large impact. The authors have made compelling graphics that help explain their techniques which significantly helps the manuscript.

My main critiques are that the many of the claims in the paper should be more qualified. While the authors have performed many experiments, there are many more that were not performed. Indeed, they can't do everything, but should rather make clear what statements are supported by their experiments. It was not clear what the limitations of their FDM printing method are based on the current manuscript. Certainly there are trade-offs that were made and the authors should discuss these limitations.

Overall a great paper.

Reviewer #2 (Remarks to the Author):

In the manuscript under consideration, the authors present a 3D printed digital material filament and blended FDM process, which enables the spatial programming of materials distribution and properties. This process makes it easy to manufacture functional gradients structures on demand.

This work is performed well, with all conclusions proven with detailed experiments. Therefore, I recommend publishing this work in this journal after the authors addressing some concerns:

1. I would suggest a minor revision of the manuscript title. The "3D Printing" and "3D Printed" seem to be a bit repetitive. The focus of the title should be Programming Functional Gradients and 3D Printed Digital Material Filament.
2. In the abstract and conclusion sections, the strength (such as tensile, bend, and the stability of multifunctional origami gripper) of the gradient material prepared in this paper compared to the traditional FDM methods should be reflected.
3. This blended FDM is of great interest. The DM filament design and preparation framework are fairly important parts of the process. The printing time of the DM filament is even much greater than that of the target object. How to optimize the forming efficiency is the main issue that the author should further consider.

Reviewer #3 (Remarks to the Author):

In this work, 'digital' blended filaments are 3D printed and then used to 3D print functional graded specimens.

Evaluation-Recommendations

1. The title is appropriate and concise.
2. The abstract accurately describes the content of the manuscript.
3. Introduction.
 - (i) In general, the text is easily read, but in some cases, the authors are not so focused when citing others' works. Please see line 58, where 12 manuscripts are cited to show that the FDM uses filaments. Please explain why use these 12 manuscripts and not the first 12 found in scholargoogle by using the following keywords: 'filament+fabrication+parameters+3d+printing'
 - (ii) The authors claimed that different materials are possible to 3D print in filament form, but mp4 files present only a simple case study of the same material with two colours (first and fourth mp4

files). In this case, the material flow can be controlled by the 3D printer when 3D prints the filament. What will happen if we use different materials? How can we control the temperatures and other 3D printing parameters? What will be the effects on 3D printed filament quality, and how can it be transferred in the final functional parts? Please discuss such issues better and cite related manuscripts about the key parameters affecting the 3D printing quality.

https://scholar.google.gr/scholar?hl=en&as_sdt=0%2C5&q=key+parameters+FFF&btnG=

4. 3D printed the filament

-The FDM process is a layer by layer process. How can different materials blend in the same layer? By using two nozzles? Please explain it better. Explain better supplementary Figure 2d.

-Two times of material switching? How was this achieved?

5. Origami part

-Explain how all the materials (I see three) are printed with the conventional and with the proposed methodology and why we have better interlayer bonding in the second one. Explain the parameters setting for each material and whether they are the same in the two approaches (in the supplementary file).

6. At first glance, the idea seems to be brilliant (3D print of digitally blended filaments), but the quality issues of the FDM process and the interlaminar bonding quality are the same in this case and maybe even more complicated. This issue should be discussed better in the manuscript.

7. Authors must show a magnified picture of the filament (cross-section and side view).

8. The text quality is good.

Please note that my feedback is only to help the authors to present sound research to the readers in a high-impact Journal.

RESPONSE TO REVIEWERS: NCOMMS-23-29800-T

General Response:

We thank the reviewers for their valuable time and effort to review our manuscript. We also sincerely appreciate the reviewers for the positive feedback and suggestions, which helped us make substantial improvement in our manuscript. In this letter, we provide point-by-point responses to the specific comments of each reviewer (which is in blue). Also listed at the end of each comment is a summary of the associated revisions we have made in the revised manuscript. The revised portions have been highlighted in the revised manuscript.

RESPONSE TO REVIEWER 1:

1. General comment

Reviewer Comment:

This work is well written, the topic is interesting and worthy of publication. The authors show how to print gradient materials with an off the shelf generic FDM printer. This could have a large impact. The authors have made compelling graphics that help explain their techniques which significantly helps the manuscript.

My main critiques are that the many of the claims in the paper should be more qualified. While the authors have performed many experiments, there are many more that were not performed. Indeed, they can't do everything, but should rather make clear what statements are supported by their experiments. It was not clear what the limitations of their FDM printing method are based on the current manuscript. Certainly there are trade-offs that were made and the authors should discuss these limitations.

Overall a great paper.

Overall response:

- We deeply appreciate the reviewer's positive feedback and constructive suggestion. Reflecting the reviewer's suggestions, we have conducted additional experiments to better support our claims. Specifically, the working mechanism of b-FDM to achieve polymer mixing through interdiffusion is more clarified with optical images and quantitative

measurements. A detailed discussion of the limitations of our work was also incorporated in the main text. Below we provide our point-by-point response to address the reviewer's comments and associated revisions.

2. Citing preceding 3D printing approaches

Reviewer Comment:

(**Introduction:** Page 2) “A widely known approach is a “digital material” printed with material jetting (MJ) 3D printing technique¹⁵. MJ process utilizes multiple inkjet printheads to simultaneously deposits numerous ink droplets. A range of material properties and its spatial gradient can be easily produced by printing a mixture of various inks in specific concentrations^{10,15-18}. However, achievable material properties from MJ are limited due to rheological constraints of the printing inks.”

- True. But we should acknowledge that tango+ to verowhite has a wide range of properties that is quite remarkable. The authors should acknowledge this rather than just mention the limitations.
- In metal printing, the LENS process is able to print gradients as well. The authors should acknowledge this.

Response:

- MJ can produce a wide range of **mechanical** properties, but we wanted to point out that it is **challenging to expand the accessible property domain beyond mechanical**, such as electrical properties, due to its nature of dispensing ink droplets. However, we admit that our current expression could potentially cause some confusion in readership. Therefore, we have revised this paragraph to better convey the limitations of MJ printing.
- Following the reviewer's second suggestion, we have added a brief comment on the titanium-based gradient alloys produced by the LENS process with appropriate reference.

Revisions:

- We have revised the Introduction in appropriate ways and added the Ref. [15] as follows.

(**Introduction:** Page 2) “MJ process utilizes multiple inkjet printheads to simultaneously deposits numerous ink droplets. A wide range of mechanical properties and its spatial gradient can be easily produced by printing a mixture of inks for rigid and soft materials in specific concentrations^{10,16-18}. However, MJ has been limited in its ability to create functional gradients in other property domains due to rheological constraints of the printing inks.”

(**Introduction:** Page 2) “Three-dimensional (3D) printing provides new opportunities for processing a wide range of materials from polymers to metal alloys to create FGMs within complex 3D geometries that would otherwise be inaccessible using traditional manufacturing^{8,14,15}.”

15. Ghanavati, R. & Naffakh-Moosavy, H. Additive manufacturing of functionally graded metallic materials: A review of experimental and numerical studies. *J. Mater. Res. Technol.* 13, 1628–1664 (2021).

3. Expression

Reviewer Comment:

(**Introduction:** Page 4) “Taking advantage of this unprecedented printing capability, we present a multifunctional origami gripper featuring rigid facets, flexible hinges, and integrated electric circuits including bending and tactile sensors, all of which were printed using a single DM filament.”

- Given the literature review, I would not call this unprecedented. Engineers have printing FGM for many years (just not in fdm).

Response:

- Our intention was to emphasize that the production of an extensive range of material properties was **achieved through our b-FDM, using only a basic, low-cost FDM printer**. We have revised the expression to avoid any potential confusion.

Revisions:

- We have revised the Introduction:
(**Introduction:** Page 4) “Taking advantage of this ~~unprecedented~~ printing capability, we present a multifunctional origami gripper featuring rigid facets, flexible hinges, and integrated electric circuits including bending and tactile sensors, all of which were printed using a basic FDM printer with a single DM filament.”

4. Terminology (fiber)

Reviewer Comment:

(**Results:** Page 6) “FDM process creates 3D objects by extruding thermoplastic filaments as a thin fiber and stacking them in a layer-by-layer fashion.”

- Fiber is not the correct word. Please change. Fiber suggest there is an actual fiber in the thermoplastic.

Response:

- We thank the reviewer for the thoughtful suggestion. To the best of our knowledge, the term “*fiber*” is also commonly used to describe the “*thin extrudate*” produced by the extrusion nozzle of the FDM printer, as used in Ref. [34] and [36]. However, to help the broad readership better understand the principle of FDM printing, we **have replaced “*fiber*” with “*extrudate*”**, which explicitly describes the extruded material.

Revisions:

- We have replaced the term ‘*fiber*’ with ‘*extrudate*’ throughout the manuscript, including the part listed below.
(**Result:** Page 6) “FDM process creates 3D objects by extruding thermoplastic filaments as a thin **extrudate** and stacking them in a layer-by-layer fashion. We exploited this to directly print a cylindrical shape of the FDM filament. A 3D printed filament, or DM filament, is designed to have a standard filament diameter of 1.75 mm, so that it can be readily fed into any FDM 3D printer for the next round of printing (**Fig 1a, Supplementary Movie 1**). With

a layer thickness of 125 μm , we stacked a total of 14 layers to print a filament (**Fig. 1b-d**). Each layer is made up of 2~4 printing lines, or **extrudates**, with a width of 440 μm each.”

5. Printing results using a DM filament and a commercial filament

Reviewer Comment:

(Results: Page 6) “Although the stair-case effect and internal voids between fibers should be compensated for with a slightly higher (~5%) filament feed rate, a DM filament and a commercial filament produced identical printing results (Supplementary Fig. 1 and Supplementary Table 1, see Supplementary Information for details).”

- Please qualify this statement. Identical based on what measurements? normally in FDM printing the consistency of the diameter is critical for long prints of high quality. Is this something that was considered?
- I don't understand how S-Table1 supports the statements. Please make it more clear.

Response:

- To clearly show the compatibility of a DM filament, we printed the tensile test specimens and a 3D complex geometry (i.e., a vase), using the same printer but with different types of filaments: a commercial filament and the DM filament. The weight, the ultimate strength, and the fracture strain of the tensile test specimens were measured. When a slightly higher feed rate (i.e., $E_f/E_{tar} = 1.05$) was used, the b-FDM printed specimens showed **99.8% in weight, 96.5% in ultimate strength, and 104.9% in fracture strain**, compared to the specimen printed with a commercial filament (See next page **Supplementary Fig. 1b-d**). As an example of a complex 3D geometry, a vase was printed with both the commercial and the DM filament. While they showed a good visual agreement, we have also measured weights and dimensions of the vases to address the reviewer's concern. The reference vase printed with the commercial filament exhibited **4.66 g in weight and 40 mm in height**. The same object printed with the DM filament exhibited **the weight of 4.63 g and the height of 39.9 mm**. We believe that it is reasonable to conclude that the results were nearly identical, considering the precision of a basic FDM printer.

- **Supplementary Table 1** provides the printing parameters used for each individual material. Addressing the reviewer’s last comment, we have revised the main text to help readers better understand the context.

Supplementary Figure 1. Extrusion ratio compensation for DM filament. **a** Schematic of the extrusion process of DM filament. In typical FDM printing with a commercially available filament, supplied amount of the filament (E_f) is set to be equal to the required amount to build the target geometry (E_{tar}). **b** Weight, **c** ultimate strength, and **d** fracture strain of printed materials using DM filaments with different extrusion ratio (E_f/E_{tar}). Red lines indicate a control group printed using commercial filaments with $E_f/E_{tar}=1$. **e** 3D object printed with commercial filament (left) and DM filament with $E_f/E_{tar}=1.05$ (right). The results show that DM filament completely replaces its commercial counterpart with proper extrusion compensation.

Revisions:

- We have revised the main text to provide a more accurate depiction of the printing process using the DM filaments.

(Results: Page 6) “To print the multimaterial DM filament, we employed slightly adjusted printing parameters such as nozzle temperature and printing speed to meet the required conditions of the materials used (**Supplementary Table 1**). Compared to commercial filaments, DM filaments have a slight geometric discontinuity such as the stair-case effect and internal voids between extrudates, which should be compensated for by adjusting the rate of material extrusion. To test the compatibility of the DM filament, tensile test specimens and a 3D

complex geometry (i.e., a vase) were printed using the same printer but with different types of filaments: a commercial filament and the DM filament (**Supplementary Fig. 1**). With a slightly higher (~5%) filament feed rate, both the DM filament and the commercial filament produced consistent results in terms of mechanical properties and dimensions (see **Supplementary Information** for details).”

- We have added an additional experimental result which confirms the compatibility of the DM filament.

(**Supplementary Information: Page 4**) “Furthermore, we printed a complex 3D geometry using commercial filament and DM filament with $E_f/E_{tar}= 1.05$ (**Supplementary Fig. 1e**). The b-FDM printed object exhibited the weight of 4.63 g and the height of 39.9 mm, which closely matches with those of the same object printed with the commercial filament having a weight of 4.66 g and a height of 40 mm. The results showed good agreement, confirming that the DM filament is comparable to commercial filaments.”

6. Polymer mixing based on interdiffusion

Reviewer Comment:

(**Results: Page 8**) “However, the mixed zone in the printed fiber is expected to be extremely small, resulting in locally aggregated materials in the final object (see **Methods** for calculations).”

- I would strongly discourage the authors from using the word “mixing” since there is no active mixing. The process is more like 2 materials side by side in a composite material.
- In the **Supplemental Fig 2**. It is not clear how some of the gradients would print. For example, near the extremes of 95% yellow and 5% cyan. Is the 5% in the middle? Can you even see the 5% cyan after extrusion?
- In **Sup Fig 2d**. The material is not homogenous throughout the DM filament. This will create a clear directional bias of the color that is visible when you print with the DM filament. This would result in the front having a different visible color (yellow) than the back (cyan) of the printed part since there is not active mixing in the nozzle. Please comment on this and provide side by side images of the front and back to show or not show this directional dependence.

(Results: Page 11) “We used PLA and polyethylene terephthalate-G (PETG) as base materials for the DM filament because they have similar mechanical properties, but do not adhere each other.”

- I had been wondering about this. This seems to indicate that the resulting FGM is more like a woven composite rather than “mixing”.

Response:

- The term “*mixing*” can be used to refer to both *convective mixing* and *diffusion mixing*. *Convective mixing* involves bulk movement or gross displacement of materials within the mixture. *Diffusion mixing* is caused by the random motion of molecules.
- The 3D printed DM filament is first in the form of discrete stacks of base materials as the reviewer described (see **Fig. 1d** and **e** below). Mixing between the base materials takes place as the DM filament passes through a heated nozzle to print the final object in the next round of 3D printing. During extrusion, *convective mixing hardly take place inside the nozzle, as the reviewer pointed out*, since there are limited bulk movement or gross displacement of materials. However, different base materials in the DM filament are still **mixed through diffusion mixing** as the DM filament pass through the extrusion nozzle, especially considering its high temperature (see **Fig. 2a** below). In order to facilitate the **diffusion-driven mixing**, we proposed a design strategy of the DM filament in which different materials are deposited in an alternating fashion (see **Fig. 2b-c** below). As a result, we successfully achieved homogeneous mixing of two polymers without any active components for *convective mixing*, as shown in **Fig. 2b-f**.
- The same principle of diffusion-driven mixing can be effectively utilized to create functional gradients (see **Fig. 3** below). The spatial design of material composition within the PLA-PETG DM filament enabled not only the precise control of the material concentration, but also **sufficient mixing of base constituents**. As a result, the b-FDM printed material gradient dramatically enhanced the interfacial bonding between PLAs and PETGs, as seen by a **three-fold increase in the fracture stress from 9.92 MPa for nongraded to 30.16 MPa for 11-step gradient profile**. Therefore, we believe that describing our strategy as “mixing” will help readers better understand the principle of b-FDM printing.

Fig. 2 b-FDM printed digital materials with tunable and multi-functional properties. **a** Schematic of the extrusion of a DM filament with two materials (illustrated in cyan and magenta). While a DM filament (diameter d_f) is supplied at a feed rate of C and passes through a heated region (length of L) in the nozzle (outlet diameter d_{out}), polymer interdiffusion (illustrated in purple) takes place across the material interface with a depth of Δx . **b** DM filament design with interdigitated multi-layer arrangements and corresponding homogeneity parameter η . **c** Schematic of different DM filament designs with varying η (top). Optical images of extrudates (bottom) were obtained by extruding DM filaments through a heated nozzle without deposition on the bed. **d** Mechanical responses of CPLA-TPU blends printed using DM filaments with varying η . **e** Elongation at break as a function of η . All error bars represent the standard deviation ($n=5$). **f** Electrical responses during stretching of specimens produced with DM filaments with varying η .

Fig. 3 b-FDM 3D printed functional gradients. **a** Design of a tensile test specimen with PLA (illustrated in red) and PETG (illustrated in light gray). The relative distance from the center is represented as x , ranging from

0 to 1. **b** Printed tensile test specimens with different material gradients at the interface. Printing direction was perpendicular to the longitudinal axis. **c** Fracture stresses and the corresponding locations of fracture from the specimens with different material gradient designs across the interface. The blue lines indicate the mean value of the fracture stresses.

- To address the reviewer’s comment, we have further analyzed the effect of mixing on the color. With two base materials (cyan and yellow) in a 1:1 ratio, we prepared a set of **DM filaments with $\eta=1$ and $\eta=11$** and printed planar specimens, as shown in **Fig. R1** below. When mixing was not considered ($\eta = 1$), directional bias was clearly observed in the specimens. Cyan and yellow were seen alternately in both the top and the bottom view, displaying a **locally aggregated, unmixed state**. Nevertheless, the macroscopic image still gives the perception of a new color (green). When the DM filament designed for better mixing ($\eta = 11$) was used, mixing between the two primary colors was facilitated during b-FDM, resulting in the production of **more uniformly blended color**, in both macroscopic and magnified views (**Fig. R1b**).

Fig. R1. Effect of homogeneous mixing. Optical images of 3D printed colors achieved by blending cyan and yellow in 1:1 ratio with (a) $\eta = 1$ and (b) $\eta = 11$.

- We have also quantitatively analyzed the color gradient shown in **Fig. 1** in order to address the reviewer’s concern on the gradient printing. To do this, we prepared a DM filament to 3D

print the same color gradient on a planar object, as illustrated in **Supplementary Fig. 2f** shown below. As designed in the previous experiments, the volume fraction of yellow was changed with stepwise variations of 8%, resulting in 11-step gradient profile between two base materials. The RGB values of each color section were measured through a commercial scanner (HP OfficeJet Pro 9010) and ImageJ, and transformed to CMYK color space using MATLAB, where C, M, Y and K denote cyan, magenta, yellow, and black, respectively. As shown in **Supplementary Fig. 2f**, both C and Y values vary according to material composition along the gradient profile, while M value, used as a reference, remained the same.

Supplementary Figure 2. f Color gradient printed in a planar object (top) and measured CMY values (bottom).

Revisions:

- Above figure has been included in Supplementary Information (**Supplementary Fig. 2f**), along with a brief discussion in the main text and the Methods.

(**Results:** Page 7) “Consequently, the DM filament is encoded with the gradually changing material composition, which is then translated into the color gradient (**Supplementary Fig. 2e and f**). The 3D printed DM filament was then used in a FDM printer to print the target object. As a result, using the DM filament composed of only two base materials, 13 levels of color gradient were precisely distributed throughout a complex 3D geometry. The result showcases the potential of b-FDM to directly fabricate FGMs.”

(Methods: Page 36) “The RGB values of each color section were calculated as the mean of the values measured across their respective sections. To plot the results on the CIE map, RGB values were converted into CIE 1931 xy color space using MATLAB function ‘rgb2xyz’. A MATLAB function ‘rgb2xyz’ was used to convert RGB values into CIE 1931 xy color space. Also, ‘makecform’ and ‘applycform’ were used for RGB to CMYK transformation, where C, M, Y, and K denote cyan, magenta, yellow, and black, respectively (Supplementary Fig. 2f).

7. Material switching during DM filament fabrication process

Reviewer Comment:

(Results: Page 9) “The boundaries between two colors are clearly visible until $\eta = 6.0$, but they become less apparent when is greater than 8.0, indicating more homogeneous mixing of the two materials.”

- This is what I was concerned about earlier. Sup Fig 2 gives a very different impression than your Figure 2 in the main paper. Please comment on the extra work of switching filaments as η increases.

Response:

- Interdigitated layer design in the DM filament improves polymer mixing, but also necessitates additional material switching during the DM filament printing process. **Supplementary Fig. 2d-f** below shows the potential to create color gradient with only a two times of material switches ($\eta = 2$ or less). On the other hand, **Fig. 2c-f** shows that increasing η up to 11 leads to significant enhancements in the mechanical and electrical behavior of b-FDM printed specimens, indicating homogeneous mixing of the base materials. However, **increasing η also requires more frequent material switching during fabrication of the DM filament**, which leads to longer printing times. We found that η of 6~8 is a suitable compromise between the sufficient mixing and the time cost.
- To clarify the difference between these two cases, we have added more explanations in the main text.

Supplementary Figure 2. DM filament design framework. **a** Voxelation of the target geometry. The target geometry is decomposed into a set of voxels with a required amount of the filament (E_i) and mapped onto the DM filament. **b** Spatial design of the material composition in the DM filament. The concentrations in the volume fraction, $\{\phi\}$, and the homogeneity parameter, η are the key design parameters to program desired material properties. **c** Constructed DM filament structure. **d** Printing schedule of the DM filament. By dividing the DM filament into three groups (denoted by $G_1 \sim G_3$), the DM filament with 13 color gradient can be printed with only a two times of material switching. **e** 3D printed DM filament in the top view (left) and side view (inset). **f** Color gradient printed in a planar object (top) and measured CMY values (bottom).

Fig. 2 b-FDM printed digital materials with tunable and multi-functional properties. **a** Schematic of the extrusion of a DM filament with two materials (illustrated in cyan and magenta). While a DM filament (diameter d_f) is supplied at a feed rate of C and passes through a heated region (length of L) in the nozzle (outlet diameter d_{out}), polymer interdiffusion (illustrated in purple) takes place across the material interface with a depth of Δx . **b** DM filament design with interdigitated multi-layer arrangements and corresponding homogeneity parameter η . **c** Schematic of different DM filament designs with varying η (top). Optical images of extrudates (bottom) were obtained by extruding DM filaments through a heated nozzle without deposition on the bed. **d** Mechanical responses of CPLA-TPU blends printed using DM filaments with varying η . **e** Elongation at break as a function of η . All error bars represent the standard deviation ($n=5$). **f** Electrical responses during stretching of specimens produced with DM filaments with varying η .

Revisions:

- We have added more explanation of material exchange process in the main text.

(Results: Page 9) “The smallest homogeneity parameter $\eta = 1$ is for the case where two materials are simply stacked without interdigitation. On the other hand, η is maximum ($\eta = 11$) when the materials change **for every layer layer by layer (13 times)** in the DM filament. **Different DM filament designs with $\eta = 3.5, 6.0,$ and 8.0 were also printed, interdigitated layer**

designs of which necessitate 4, 7, and 9 times of manual material switching during the filament fabrication process, respectively.”

(**Results:** Page 10) “As η gradually increased, electrical conductivity was maintained at higher strains up to 69%. Therefore, CPLA-TPU blends can be used for stretchable sensing applications. Similar results were also obtained for different CPLA-TPU mixing ratios (**Supplementary Fig. 4**). The results demonstrate that increasing η facilitates effective blending of basic constituents, resulting in an enhanced performance of b-FDM printed digital materials. However, this increase in η also requires more frequent switch of the feedstock throughout the fabrication of the DM filament, resulting in a time-consuming process (approximately 3 minutes for each change). Through our investigation, we have identified a suitable compromise in this trade-off at values of η ranging from 6 to 8.”

8. Tool path design and evaluation methods for mechanical gradient designs

Reviewer Comment:

(**Results:** Page 12) “As the number of gradient steps increased, there was a three-fold enhancement in the average fracture stress from 9.92 MPa for nongraded to 30.16 MPa for 11-step.”

- How did the tool path of the print head change for these prints. Please provide evidence that the tool path had no or little effect since it is well known that the tool path can significantly affect the strength of a part.

(**Results:** Page 12) “In addition, increasingly widespread distribution of fraction locations across the specimen was observed as compositions changed more gradually.”

- Fracture location does not seem like the best metric since fracture has many contributions. Using DIC and tracking the strain would be more convincing.
- Please comment on how fast the gradient can change and still achieve these properties. For example, does the transition region need to be 2 times the height?

Response:

- We thank the reviewer for insightful questions. As illustrated in **Fig. 3b** and Methods, all specimens were fabricated with **the tool path in the transverse direction** (or perpendicular to the specimen's longitudinal axis) (i.e., x -axis). We intentionally chose this print direction because specimens printed in the transverse direction are most susceptible to forces applied during the tensile test. Our study clearly shows in **Fig. 3c** that a material gradient at the interface dramatically enhances the bonding strength. The average fracture stress of the 11-step specimen was measured as 30.16 MPa, which is about **3-fold improvement** compared to that of the nongraded specimen. Moreover, it reaches approximately 75% of that of the reference specimens (the fracture stress of specimens consisting only of PLAs or PETGs are 40 MPa or 44 MPa, respectively). For better understanding, we relocated the detailed explanation from Methods to Results.

Fig. 3 b-FDM 3D printed functional gradients. **a** Design of a tensile test specimen with PLA (illustrated in red) and PETG (illustrated in light gray). The relative distance from the center is represented as x , ranging from 0 to 1. **b** Printed tensile test specimens with different material gradients at the interface. Printing direction was perpendicular to the longitudinal axis. **c** Fracture stresses and the corresponding locations of fracture from the specimens with different material gradient designs across the interface. The blue lines indicate the mean value of the fracture stresses.

- The fracture location typically appears randomly in a homogeneous material. In multimaterial 3D printing, however, fractures predominantly occur at the interfaces between different materials. This was also observed in other works such as Ref. [16], [21], and [34]. Therefore, we believe that **the wide distribution of fracture locations as well as the enhanced fracture**

stress in the graded specimens can serve as compelling evidence that the material gradient design has improved weak interfacial bonding.

- We thank the reviewer for the suggestion of using DIC. However, tensile modulus of same specimens printed with PLAs and PETGs are 1.06 GPa and 0.97 GPa, respectively, which would result in similar strain fields within the specimens regardless of material distribution. The focus of our experiment was on how a material gradient helps **to improve the fracture strength and to evenly distribute fracture locations** as it would be seen from a single material specimen.
- Regarding the reviewer's last comment, the spatial resolution of the gradient is determined by the geometry of the specimen and the programming resolution of b-FDM. It is challenging to program multiple material properties within a narrow range as **the printing resolution of basic FDM is constrained by the nozzle size (~0.4 mm)**. Consequently, programming a functional gradient would require a broad interface between materials. For instance, 11-step gradient profile illustrated in **Fig. 3** has a total band size of 10 mm along the *x*-axis. This is threefold the thickness of the specimen (3.2 mm), which is designed to provide sufficient bonding between the vertically stacked extrudates. While a more distinctive interface with a narrower band size can be achieved with a more abrupt change in the material composition (i.e., 1-step or 5-step profile), it may lead to a loss of the characteristics of FGM, resulting in a limited enhancement of bonding strength (**Fig. 3c**).

Revisions:

- We have **moved the related explanation from Methods to Results**. Detailed discussions on the gradient profile designs have also been added.

(**Methods:** Page 35) “The total gauge length was set to be 48 mm to contain the second materials, PETGs, with length of 8 mm, and material gradients at the interfaces with length of 10 mm placed at both sides, symmetrically. To create different gradient profiles, we first designed the DM filament with 11-step gradient profile (**Supplementary Fig. 7**). Different DM filament designs were selected to create other gradient profiles; specifically, 50/50 was used for 1-step profile, while 16/84, 35/65, 50/50, 65/35, 84/16 for 5-step profile. **All**

specimens were printed with the printing direction (movement of the nozzle) oriented perpendicular to the specimen's longitudinal axis (i.e. x direction), since FDM printed objects are most susceptible to forces applied in this direction (Fig. 3b).”

(Results: Page 12) “In the tensile test specimens, we positioned the two materials in a symmetric arrangement, with PLA (illustrated in red) at the center and PETG (illustrated in gray) at both ends, as shown in Fig. 3a and b (See Methods for details). x represents the relative distance from the center, ranging from 0 to 1. All specimens were printed with the nozzle path in the transverse direction (or perpendicular to the specimen's longitudinal direction, i.e., x -axis), since FDM printed objects are most susceptible to forces applied in this direction (Fig. 3b). To investigate the effect of material gradient on the interfacial bonding, we prepared three PLA-PETG DM filaments to create different profiles of material gradient. The volume fraction of PETG was changed with stepwise variations of 50%, 16%, and 8%, resulting in 1-, 5-, and 11-step gradient profiles within the test specimens, respectively (Fig. 3c, bottom). The interdigitated layer arrangements in the DM filaments were designed to ensure uniform mixing of PLA and PETG during b-FDM process (Supplementary Fig. 7). In addition, to serve as a control experiment, a specimen with a sharp material change (nongraded) was also printed (see Methods). Fracture stresses and corresponding fracture locations obtained from tensile tests are shown in Fig. 3c. As the number of gradient steps increased, there was a three-fold enhancement in the average fracture stress ($\sigma_{\text{fracture, avg}}$) from 9.92 MPa for nongraded to 30.16 MPa for 11-step. In addition, increasingly widespread distribution of fraction locations across the specimen was observed as compositions changed more gradually. It is noteworthy that there is a trade-off between the steepness of the material gradient and the band size of the material interface. A more abrupt change in the material composition (i.e., 1-step or 5-step profile) would result in a more distinctive, narrower interface; however, it may also lead to a loss of the characteristics of FGM, resulting in a limited enhancement of bonding strength. This result suggests that b-FDM-enabled material gradient programming can facilitate seamless multimaterial 3D printing and promote robust bonding between different materials with mechanically invisible material interfaces.”

9. Joule heating of the conductive letters

Reviewer Comment:

(Results: Page 14) “Due to the presence of embedded CB nanoparticles, the printed letters are visually indistinguishable because CPLA, TPU33, and TPU66 all appear black in color (Fig. 3h, inset). However, the application of a DC voltage of 85V to each printed letter revealed the letters in the thermal image captured by an infrared camera, due to different levels of Joule heating and subsequent thermal gradient.”

- This demo is fun and interesting but lacks scientific rigor. I would suggest adding more quantitative results for the joule heating.

Response:

- We appreciate the reviewer’s suggestion for improving this paper. We fully agree that our demonstration should be further analyzed in the scientific point of view. Therefore, we performed an additional analysis with the Joule-heating capacity of the specimen. The resistance of each printed letter was measured as 3.26, 7.81, and 25.3 k Ω , respectively. When applying a voltage of 85V in parallel, the **Joule heating powers (P)** are expected to be 2.21, 0.93, and 0.29 W, respectively, based on $P=V^2/R$, where V and R denote the input voltage and the resistance, respectively (**Supplementary Fig. 8b**). Assuming the same specific heat, temperature increase ΔT from room temperature (25 °C) were 31.2, 14.2, and 4.9 °C, respectively, which are proportional to the supplied power. This result shows that thermal gradients can also be precisely programmed through the material design of a DM filament.

Supplementary Figure 8b. b-FDM printing of different levels of electrical resistance onto the specific printing pattern and circuit design for Joule heating.

Revisions:

- Above figure has been included in Supplementary Information (**Supplementary Fig. 8b**), along with a brief discussion in the main text.

(**Results:** Page 13-14) “Based on this, we designed a DM filament with distinct segments having different electrical resistance, 3.26, 7.81, and 25.3 k Ω , respectively (**Supplementary Fig. 8b**). Moreover, we ensured that the longitudinal position of each segment in the DM filament was synchronized with the printing sequence, so that each segment was printed in the correct order and accurately mapped onto the specific printing pattern (**Supplementary Fig. 8c**). Specifically, the letters ‘S’, ‘N’, and ‘U’ were printed with CPLA, TPU33, and TPU66, respectively, where the number denotes the concentration of TPU (**Fig. 3g**). The substrate was printed with pristine TPU for electrical insulation. Due to the presence of embedded CB nanoparticles, the printed letters are visually indistinguishable because CPLA, TPU33, and TPU66 all appear black in color (**Fig. 3h**, inset). However, the application of a DC voltage of 85V in parallel induces a different level of Joule heating power, P , of 2.21, 0.93, and 0.29 W, respectively (based on $P=V^2/R$, where V and R denote the input voltage and the resistance, respectively). As a result, three letters were heated above room temperature (25 °C) with the increase proportional to the supplied power (31.2, 14.2, and 4.9 °C, respectively), making the invisible letters appear in the infrared image (**Fig. 3h**, see Materials and Methods).”

10. Tactile sensing in the griper

Reviewer Comment:

(**Results:** Page 16) “Upon contact, the tactile sensors conformed to the object to provide a secure grip and produced a noticeable change in the signal.”

- In the figure it looks like it is picking up a metal object. Is the metal touching the electrical circuit?
- Rather than plotting the %changeR as a function of time in Fig 4h ii, it would be more helpful to have a force sensor and show the force value on the x-axis.

Response:

- We thank the reviewer for insightful comments. As shown in **Fig. 4a**, the integrated electrical circuit in the gripper was printed on **the TPU layers, which serve not only as the flexible hinges, but also as an insulator**. Therefore, the integrated sensors remain electrically isolated from the object while grasping it. The change in the signal (**Fig. 4h, ii**) is attributed to the large deformation of the tactile sensors as they conform to the object.

Fig. 4 b-FDM 3D printed multifunctional origami gripper. a Schematic illustration of the multimaterial, multifunctional origami gripper.

Fig. 4 b-FDM 3D printed multifunctional origami gripper. h Resistance variation of the folding sensor and the tactile sensor during the sequential folding and contact.

- To our best knowledge, measuring the changes in resistance, $\Delta R/R_0$, to characterize the response of the sensors is a widely accepted approach, as demonstrated in Ref. [38] or other preceding literature listed below. While we can convert $\Delta R/R_0$ into contact force through a simple calibration as the reviewer suggested, the strength of the b-FDM that we aim to emphasize lies in **its capability to systematically integrate independent functional electric circuits using only a basic FDM printer**. Therefore, we believe that directly presenting the attained electric property ($\Delta R/R_0$) more effectively fulfills the objective of validating the underlying concept.

50. Maurya, D. et al. 3D printed graphene-based self-powered strain sensors for smart tires in autonomous vehicles. *Nat. Commun.* **11**, 5392 (2020).

51. Wang, Z. et al. Full 3D Printing of Stretchable Piezoresistive Sensor with Hierarchical Porosity and Multimodulus Architecture. *Adv. Funct. Mater.* **29**, 1807569 (2019).

52. Kotikian, A. et al. Innervated, Self-Sensing Liquid Crystal Elastomer Actuators with Closed Loop Control. *Adv. Mater.* **33**, 2101814 (2021).

53. Valentine, A. D. et al. Hybrid 3D Printing of Soft Electronics. *Adv. Mater.* **29**, 1703817 (2017).

Revisions:

- The above references have been included and cited in the main text.

(**Results:** Page 16) “The folding sensor and tactile sensors were individually connected to the ohmmeter in order to **characterize the responses with** changes in resistance, $\Delta R/R_0$ (**Supplementary Fig. 9**)^{38,50-53}.”

11. Limitations of the work

Reviewer Comment:

(**Conclusion:** Page 17) “It is also worth highlighting that the presented approach holds universal applicability, as the DM filaments can be 3D printed and utilized with any FDM 3D printer.”

- You seem limited by how long of a DM filament you can print at once. There is limited length of material you could print at once. Please make note of this in the paper.

- This paper has primarily focused on printing and testing in 2.5D like shapes where the gradient is in the XY plane. The color gradient vase shows that you can print gradients in the Z, but this did not test mechanical properties. Please make note of this and/or redo experiments to evaluate properties in Z direction gradient.

Response:

- We thank the reviewer for identifying these issues. We fully agree that the limitations of our approach should be thoroughly discussed in the manuscript. As pointed out by the reviewer, with the particular printer we utilized in this study, **the length of the DM filament that can be printed at once is limited to about 20 m**. It could be improved by using a printer with a larger build size, which aligns with our future work.
- As suggested by the reviewer, applying functional gradients in Z direction could be an intriguing approach that can leverage the capability of 3D printing. In general, 3D printed objects are vulnerable to forces applied in the stacked direction (i.e., z-axis). This is similar to their vulnerability to forces acting perpendicular to the printing direction on the XY plane (**Fig. 3b**). As shown in **Fig. 3a-c** below, we have demonstrated the efficacy of the b-FDM printed gradient design in improving the interfacial bonding between dissimilar materials arranged in the XY plane. Thus, we anticipate that **the same principle of programming functional gradients can be readily applied to the z-axis**. This hypothesis is also supported by the robust bonding between rigid and flexible materials that are vertically stacked in the b-FDM printed origami gripper (**Fig. 4e and f, Supplementary Movie 2**).

Fig. 3 b-FDM 3D printed functional gradients. **a** Design of a tensile test specimen with PLA (illustrated in red) and PETG (illustrated in light gray). The relative distance from the center is represented as x , ranging from

0 to 1. **b** Printed tensile test specimens with different material gradients at the interface. Printing direction was perpendicular to the longitudinal axis. **c** Fracture stresses and the corresponding locations of fracture from the specimens with different material gradient designs across the interface. The blue lines indicate the mean value of the fracture stresses.

Fig. 4 b-FDM 3D printed multifunctional origami gripper. e-f The effect of the mechanical gradient on the integration of heterogeneous properties. The folding behavior of **e** the b-FDM printed origami gripper is contrasted with **f** the conventional FDM printed gripper which exhibits clear delamination at the interface.

Revisions:

- We have revised the Conclusion by incorporating a detailed discussion of the limitations of our work.

(**Conclusion:** Page 17) “Hence, the b-FDM with 3D printed DM filaments offers a new pathway to unleash the full potential of FGMs for various engineering application. However, the challenges may arise when attempting to program various material properties within a narrow range, primarily due to the limited printing resolution of the standard FDM⁵⁴. Consequently, achieving functional gradients through b-FDM programming may require a wide material interface. In addition, there is also room for improvement in optimizing printing time for DM filaments and extending the length of a printable DM filament in a single printing. Nevertheless, it is worth highlighting that the presented approach holds universal applicability, as the DM filaments can be 3D printed and utilized with any FDM 3D printer (**Supplementary Movie 4** and **Supplementary Fig. 10**). Thus, we envision that the presented method can readily serve as a powerful add-on to existing FDM printers that are already in use worldwide, pushing the boundaries of what is achievable in 3D printing.”

RESPONSE TO REVIEWER 2:

1. General comment

Reviewer Comment:

- In the manuscript under consideration, the authors present a 3D printed digital material filament and blended FDM process, which enables the spatial programming of materials distribution and properties. This process makes it easy to manufacture functional gradients structures on demand. This work is performed well, with all conclusions proven with detailed experiments. Therefore, I recommend publishing this work in this journal after the authors addressing some concerns:

Response:

- We deeply appreciate the reviewer's valuable time and effort for evaluating this manuscript. The following is the point-by-point responses to the reviewer's comments and suggestions, and appropriate modifications were reflected to the revised version of the manuscript.

2. Suggested improvements for the title

Reviewer Comment:

- I would suggest a minor revision of the manuscript title. The "3D Printing" and "3D Printed" seem to be a bit repetitive. The focus of the title should be Programming Functional Gradients and 3D Printed Digital Material Filament.

Response:

- We appreciate the reviewer for the thoughtful suggestion. We believe that the originality of our work lies in **the utilization of 3D printing to fabricate a filament that, in turn, facilitates the creation functional gradients through 3D printing**. In order to underscore this distinctive aspect, we composed the title, "*3D Printing with a 3D Printed Digital Material Filament for Programming Functional Gradients.*" **We believe that the repetition of the term "3D printing" in the title serves to increase the visibility of this work and capture the attention of readership.** Thus, we would like to keep this title unchanged.

3. Reflecting the strengths of the work in the Abstract and Conclusion

Reviewer Comment:

- In the abstract and conclusion sections, the strength (such as tensile, bend, and the stability of multifunctional origami gripper) of the gradient material prepared in this paper compared to the traditional FDM methods should be reflected.

Response:

- We thank the reviewer's constructive suggestion. We agree that the advantages and benefits of our new fabrication strategy could be better emphasized by comparing with the traditional FDMs, which lacks the capability to fabricate functional gradients.

Revisions:

- We have revised the abstract and conclusion in alignment with the reviewer's suggestions.

(**Abstract:** Page 1) “This blended FDM (b-FDM) printing enables spatial programming of material properties in extreme variations, including mechanical strength, electrical conductivity, and color, which are otherwise impossible to achieve with traditional FDMs. The b-FDM with DM filaments can be readily adopted to any standard FDM printer, enabling low-cost production of FGMs.”

(**Conclusion:** Page 17) “Decoding of this information through the b-FDM process enables physical realization of both the intended geometry and the desired property distribution in the final printout. To demonstrate the unique advantages of our method, we presented various applications, including strain sensor, multicolor printout, electrical resistance distributions, and functional origami gripper, all of which are beyond the capabilities of conventional FDMs.”

4. Time cost for printing DM filament

Reviewer Comment:

- This blended FDM is of great interest. The DM filament design and preparation framework are fairly important parts of the process. The printing time of the DM filament is even much greater than that of the target object. How to optimize the forming efficiency is the main issue that the author should further consider.

Response:

- We agree with the reviewer that the forming efficiency of the DM filament is an important factor to consider. We anticipate that the printing speed for DM filament fabrication can be improved with further optimization of printing process parameters. Also, our ongoing endeavor includes increasing the printable length of the DM filament using a custom-designed apparatus, which will help to speed up the overall process cycle. While we acknowledge the additional time cost for the DM filament production, we would like to put a strong emphasis on the fact that our b-FDM approach has enabled **the direct, low-cost 3D printing of multiple material properties over a variety of domains, which is otherwise impossible to achieve with conventional FDM.**

Revisions:

- We have added a brief discussion of the time cost for fabricating the DM filament in the conclusion.

(Conclusion: Page 17) “In addition, there is also room for improvement in optimizing printing time for DM filaments and extending the length of a printable DM filament in a single printing. Nevertheless, it is worth highlighting that the presented approach holds universal applicability, as the DM filaments can be 3D printed and utilized with any FDM 3D printer (Supplementary Movie 4 and Supplementary Fig. 10).”

RESPONSE TO REVIEWER 3:

1. Evaluations

Reviewer Comment:

- In this work, ‘digital’ blended filaments are 3D printed and then used to 3D print functional graded specimens.

Evaluation-Recommendations

- 1. The title is appropriate and concise.
- 2. The abstract accurately describes the content of the manuscript.
- 8. The text quality is good.

Response:

- We thank the reviewer for the time and effort for reviewing this manuscript and for the positive feedback. The following is the point-by-point responses to the reviewer’s other comments and suggestions.

2. Citations

Reviewer Comment:

- 3. (i) In general, the text is easily read, but in some cases, the authors are not so focused when citing others' works. Please see line 58, where 12 manuscripts are cited to show that the FDM uses filaments. Please explain why use these 12 manuscripts and not the first 12 found in scholargoogle by using the following keywords:
'filament+fabrication+parameters+3d+printing'

Response:

- We thank the reviewer for pointing this out. We agree that the preceding literature could be better presented in the manuscript with clear purpose.

Revisions:

- Among the many pieces of preceding works that we found through the suggested search keywords, **we have incorporated relevant ones listed below.**

(**Introduction:** Page 3) “However, FDM printing of functional gradients remains challenging since typical FDMs lack the capability for precise spatial control of material composition. This limitation stems from the fact that the extrusion nozzle in FDM can process only a single material at a time in the form of a filament^{32,33}.”

32. Hamat, S. et al. Influence of filament fabrication parameter on tensile strength and filament size of 3D printing PLA-3D850. *Mater. Today: Proc.* 74, 457–461 (2023).

33. Shaqour, B. et al. Gaining a better understanding of the extrusion process in fused filament fabrication 3D printing: a review. *Int. J. Adv. Manuf. Technol.* 114, 1279–1291 (2021).

3. Parameter tuning for FDM printing quality

Reviewer Comment:

- 3. (ii) The authors claimed that different materials are possible to 3D print in filament form, but mp4 files present only a simple case study of the same material with two colours (first and fourth mp4 files). In this case, the material flow can be controlled by the 3D printer when 3D prints the filament. What will happen if we use different materials? How can we control the temperatures and other 3D printing parameters? What will be the effects on 3D printed filament quality, and how can it be transferred in the final functional parts? Please discuss such issues better and cite related manuscripts about the key parameters affecting the 3D printing quality.

https://scholar.google.gr/scholar?hl=en&as_sdt=0%2C5&q=key+parameters+FFF&btnG=

- 6. At first glance, the idea seems to be brilliant (3D print of digitally blended filaments), but the quality issues of the FDM process and the interlaminar bonding quality are the same in this case and maybe even more complicated. This issue should be discussed better in the manuscript.

Response:

- We thank the reviewer for the insightful question. It is correct that the DM filaments used in the examples shown in **Supplementary Videos 1 and 4** were printed with the same material (PLA) with two different colors. However, other materials that are widely used for FDM, such as ABS, TPU, and PETG, also have similar melting temperatures, which warrants the compatibility of different filament materials in FDM (See below **Supplementary Table 1** for optimal printing parameters). In fact, we actually employed those materials to print the DM filaments for other examples in the manuscript, including the strain sensor (**Fig. 2**, TPU-CPLA), the tensile test specimens (**Fig. 3**, PLA-PETG), and the origami gripper (**Fig. 4**, TPU, PLA, CPLA). For printing the multimaterial DM filaments, we only had to **slightly adjust the printing parameters such as nozzle temperature and printing speed to meet the required conditions for the materials used** (See below **Supplementary Table 2**). Once fed into the heated nozzle for printing the final object, different materials in the DM filament melted and mixed during extrusion, as discussed in **Fig. 2**. No critical issue was found in the multimaterial DM filaments, as well as in the final objects printed with them.
- In terms of **the quality of a 3D printed filament**, we precisely designed the geometry of the DM filament considering the precision and quality of FDM printing. Compared to commercial filaments, minor defects such as internal voids and the stair-case effect were observed in the 3D printed filament, which may affect the mechanical properties of the final printout. In order to compensate this, we employed a slightly increased extrusion rate (i.e., $E_f/E_{tar}=1.05$) when a DM filament was printed. As a result, the b-FDM printed specimens showed **99.8% in weight, 96.5% in ultimate strength, and 104.9% in fracture strain**, compared to the specimen printed with a commercial filament, as shown in **Supplementary Fig. 1b-d** below. As the reviewer pointed out, parameter control is very important to ensure print quality in FDM printing. However, we want to emphasize that the strength of our b-FDM printing lies in the fact that it enables **direct 3D printing of multiple material properties with precise spatial control**, which cannot be accomplished otherwise with conventional FDM.

- Following the reviewer’s constructive suggestion, we have added more details on the experimental conditions including the current printing parameters we used for stable printing. We have also added discussions on further improvements to Conclusion.

Supplementary Figure 1. Extrusion ratio compensation for DM filament. **a** Schematic of the extrusion process of DM filament. In typical FDM printing with a commercially available filament, supplied amount of the filament (E_f) is set to be equal to the required amount to build the target geometry (E_{tar}). **b** Weight, **c** ultimate strength, and **d** fracture strain of printed materials using DM filaments with different extrusion ratio (E_f/E_{tar}). Red lines indicate a control group printed using commercial filaments with $E_f/E_{tar}=1$.

Revisions:

- We have revised the Methods to provide more detailed explanations of the printing parameters. We also have noticed that **Supplementary Table 1** and **2** may misinform readers; thus, we revised them to ensure clarity.

(Methods: Page 33) “The printing parameters including the temperature and the printing speed were adjusted for stable printing (**Supplementary Table 1** and **2**). To ensure good adhesion, the print bed was heated to 70 °C, and the nozzle temperature was changed from 220 °C to 240 °C, following the manufacture’s recommendation. When extruding DM filaments, the nozzle temperature was set to be the maximum allowable temperature to ensure sufficient material supply. For instance, we used the nozzle temperature of 220 and 240 °C accordingly to build the PLA-PETG DM filaments shown in **Fig. 3**, but the temperature was fixed to 240 °C while 3D printing the final specimen. Other parameters including the printing time of each DM filament and final object can be found in **Supplementary Table 2.**”

Supplementary Table 1: 3D printing parameters to build the DM filaments.

Feedstocks	PLA	PETG	CPLA	TPU
Nozzle temperature (°C)	220	240	220	235
Bed temperature (°C)	70	70	70	70
Printing speed (mm/s)	40	40	40	30
Printing speed at first layer (mm/s)	20	20	20	15
Using cooling fan	On	On	On	Off
Extrusion ratio (E_f/E_{tar})	1.05	1.05	1.05	1.10

Supplementary Table 2: 3D printing parameters for 3D printing of target objects using DM filaments.

Items	Nozzle temperature (°C)	Printing speed (mm/s)	Extrusion ratio (E_f/E_{tar})	Filament length (m)	DM filament printing time (h)	Target object printing time (h)
3D vase with color gradient (Fig. 1)	220	40	1.05	1.63	1.5	0.8
Strain sensor (Fig. 2)	235	30	1.1	0.66	<0.5	<0.5
Mechanically graded specimen (Fig. 3)	240	40	1.05	1.76	Depends on the type (<2)	1
Color gradient (Fig. 3)	220	40	1.05	2.18	2	0.5
Origami gripper (Fig. 4)	235	30	1.1	20.03	6	2.5
‘SNU’ with thermal gradient (Supplementary Fig. 8)	235	30	1.1	2.69	2.5	0.8

- We have included the additional reference obtained from the search keywords suggested by the reviewer, along with discussions of the limitations and future improvements of our work.

54. Kechagias, J., Chaidas, D., Vidakis, N., Salonitis, K. & Vaxevanidis, N. M. Key parameters controlling surface quality and dimensional accuracy: a critical review of FFF process. *Mater. Manuf. Process.* **37**, 963–984 (2022).

(Conclusion: Page 17) “However, the challenges may arise when attempting to program various material properties within a narrow range, primarily due to the limited printing resolution of the standard FDM⁵⁴. Consequently, achieving functional gradients through b-FDM programming may require a wide material interface. In addition, there is also room for improvement in optimizing printing time for DM filaments and extending the length of a printable DM filament in a single printing.”

4. 3D printing DM filaments

Reviewer Comment:

- 4. The FDM process is a layer by layer process. How can different materials blend in the same layer? By using two nozzles? Please explain it better. Explain better supplementary Figure 2d.
- Two times of material switching? How was this achieved?

Response:

- Let us take **Supplementary Fig. 2d** below as an example. For printing the DM filament, two base filaments (cyan and yellow in the case) were used. A DM filament is formed by stacking 14 printed layers. Since a standard FDM printer usually has **only a single nozzle**, base material would be switched every layer, which would cause too much interruption during multimaterial DM filament printing. However, through proper design of the material arrangement in the DM filament, we were able to optimize the printing sequence to minimize the number of material switch. For example, after depositing yellow material first (**Supplementary Fig. 2d, i**), we paused the printing to replace the yellow filament with the cyan filament (**1st switch**). Then, we resumed the printing to print the cyan part on top of the yellow part (**Supplementary Fig. 2d, ii**). Then, we switched the cyan filament back to the yellow (**2nd switch**) and continued

the printing to complete the remaining part of the DM filament (**Supplementary Fig. 2d, iii**). As a result, the concentrations of yellow and cyan vary along the DM filament (**Supplementary Fig. 2b**). Therefore, when this DM filament was supplied for printing, mixing between the two base materials takes place to create the prescribed color gradient (13 levels of color gradient) in the final object (**Supplementary Fig. 2a**).

Supplementary Figure 2. DM filament design framework. **a** Voxelation of the target geometry. The target geometry is decomposed into a set of voxels with a required amount of the filament (E_i) and mapped onto the DM filament. **b** Spatial design of the material composition in the DM filament. The concentrations in the volume fraction, $\{\phi\}$, and the homogeneity parameter, η are the key design parameters to program desired material properties. **c** Constructed DM filament structure. **d** Printing schedule of the DM filament. By dividing the DM filament into three groups (denoted by $G_1 \sim G_3$), the DM filament with 13 color gradient can be printed with only two times of material switching.

5. 3D printed origami gripper

Reviewer Comment:

- 5. (Origami part) Explain how all the materials (I see three) are printed with the conventional and with the proposed methodology and why we have better interlayer bonding in the second one.
- Explain the parameters setting for each material and whether they are the same in the two approaches (in the supplementary file).

Response:

- Following the reviewer's suggestion, the multifunctional origami gripper was replicated with the conventional FDM printer for comparison. Three base materials, namely, PLA, CPLA, and TPU filaments, were used as the reviewer described. The flexible hinge was firstly printed using pristine TPUs, followed by a material switch from TPUs to CPLAs to print the integrated circuit, and subsequent switching and deposition of PLAs onto it to create rigid facets. Since interfacial bonding between different materials is weak, the conventional FDM-printed origami gripper exhibited **clear delamination as soon as it was folded (Supplementary Movie 2)**. In contrast, b-FDM printing with a single DM filament which contains all necessary material properties enables **sharp folding with robust interfacial bonding** in the gripper. This approach aligns with a well-accepted concept of improving interfacial bonding through functional gradient, as seen by Ref. [3], [8], and [21]. The quantitative results obtained with the b-FDM printed mechanical gradients between PLAs and PETGs further support our claim, as shown in **Fig. 3**.
- As for the printing parameters, we employed equivalent values as indicated in **Supplementary Table 1 and 2** for both conventional printing and b-FDM printing, except for the slightly increased extrusion rate that is specifically required for extruding the DM filament. For better understanding, we have included a brief description in the Result.

Revisions:

- We have added a brief description on the conventional FDM printing of the origami gripper. (**Results:** Page 15-16) “The individual components of the gripper and their corresponding digital materials were encoded onto specific locations on the DM filament, considering the printing sequence of the whole device. Specifically, the filament was programmed to create the flexible hinges as a substrate (**Fig. 4d, i**), the electrical components, the mechanical gradient, and the rigid facets, in sequential order across the layers (**Fig. 4d, ii-iv**). **Meanwhile, for comparison, we replicated the gripper by manually switching pristine TPU, CPLA, and PLA filaments to create the hinges, the electrical components, and the rigid facets, respectively.**”

6. Adding picture of DM filament

Reviewer Comment:

- 7. Authors must show a magnified picture of the filament (cross-section and side view).

Response:

- We agree that the magnified picture of the filament will help readers better understand how the DM filament is fabricated. Since the cross-section of the DM filament is already presented in **Fig. 1d**, we have captured additional images to illustrate the side view in detail.

Revisions:

- We have added **Supplementary Fig. 2e** in Supporting Information.

Supplementary Figure 2. e 3D printed DM filament in the top view (left) and side view (inset).

REVIEWERS' COMMENTS

Reviewer #2 (Remarks to the Author):

All comments were addressed and the manuscript was adjusted accordingly. From my point of view, the manuscript can be published in the journal Nature Communications.

Reviewer #3 (Remarks to the Author):

The authors have followed all my recommendations, and I am now suggesting this manuscript for publication.